# Evidential Reasoning Advances Interpretable Real-World Disease Screening

**Chenyu Lian** [1 2]  **Hong-Yu Zhou** [3]  **Jing Qin** [1 2]

## Abstract

Disease screening is critical for early detection and timely intervention in clinical practice. However, most current screening models for medical images suffer from limited interpretability and suboptimal performance. They often lack effective mechanisms to reference historical cases or provide transparent reasoning pathways. To address these challenges, we introduce `EviScreen`, an evidential reasoning framework for disease screening that leverages region-level evidence from historical cases. The proposed `EviScreen` offers retrospection interpretability through regional evidence retrieved from dual knowledge banks. Using this evidential mechanism, the subsequent evidence-aware reasoning module makes predictions using both the current case and evidence from historical cases, thereby enhancing disease screening performance. Furthermore, rather than relying on post-hoc saliency maps, `EviScreen` enhances localization interpretability by leveraging abnormality maps derived from contrastive retrieval. Our method achieves superior performance on our carefully established benchmarks for real-world disease screening, yielding notably higher specificity at clinical-level recall. Code is publicly available at https://github.com/DopamineLcy/EviScreen.

## 1. Introduction

Medical imaging serves as a crucial tool for disease screening, as it provides visual clues that help clinicians locate

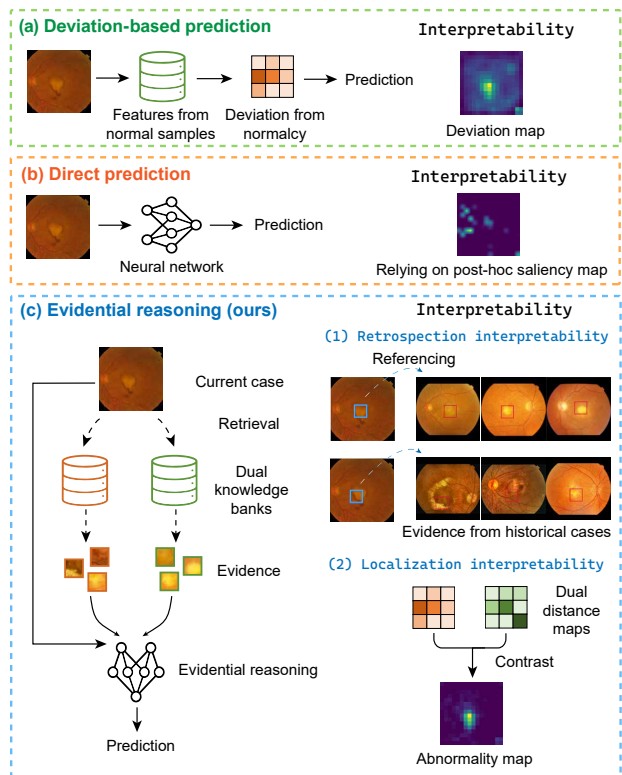

*Figure 1.* Comparison of three paradigms for disease screening. (a) Deviation-based prediction methods only generate deviation maps to provide localization interpretability. (b) Direct prediction methods rely on post-hoc saliency maps, such as Grad-CAM, to achieve localization interpretability. (c) We propose **evidential reasoning**, which retrieves regional evidence from historical cases. Our `EviScreen` not only provides *retrospection interpretability*, mirroring the decision-making process of clinicians, but also produces better *localization interpretability* than deviation-based approaches through more focused abnormality maps.

potential anomalies (Adams et al., 2023; Zhang et al., 2024; Aggarwal et al., 2021). In clinical practice, specialists usually rely on a combination of their professional experience and *evidence from historical cases* to make clinical judgments (Fanaroff et al., 2019; Sackett et al., 1996). However, mainstream disease-screening pipelines lack this critical ability to trace back to historical cases, reducing their interpretability, trustworthiness, and performance. Although saliency map-based methods such as Grad-CAM (Selvaraju et al., 2017) can offer some localization interpretability,

---
[1]The Center for Smart Health, School of Nursing, the Hong Kong Polytechnic University, Hong Kong, China [2]Research Institute for Smart Ageing, the Hong Kong Polytechnic University, Hong Kong, China [3]School of Biomedical Engineering, Tsinghua Medicine, Tsinghua University, Beijing, China. Correspondence to: Hong-Yu Zhou, Jing Qin <hongyu.zhou.ai@gmail.com, harry.qin@polyu.edu.hk>.

*Proceedings of the 43$^{rd}$ International Conference on Machine Learning*, Seoul, South Korea. PMLR 306, 2026. Copyright 2026 by the author(s).

their quality is still considered unsatisfactory (Saporta et al., 2022). More importantly, these approaches fail to provide an evidence-based reasoning process for their predictions, leading to a lack of retrospection interpretability and potentially constraining their performance.

There are two major paradigms for disease screening. The first paradigm, deviation-based prediction (Figure 1a), aims to solve the problem by one-class classification (also known as unsupervised anomaly detection), which learns a mode of normalcy using normal cases (Roth et al., 2022; Li et al., 2025; Guo et al., 2023; Lian et al., 2025). While this allows for the generation of deviation maps by identifying deviations of the current case from normalcy, a key limitation is that it cannot fully utilize information from positive cases. This omission can impair performance, particularly with complex modalities such as chest X-rays and dermoscopic images. Another intuitive approach is direct prediction (Figure 1b) by fully supervised classification, which frames the task as a binary problem, training on both normal and pathological images (Zhang et al., 2023; Cai et al., 2024b; Adams et al., 2023). The interpretability of these models typically relies on post-hoc saliency maps that visualize regions deemed important for the prediction (Selvaraju et al., 2017; Marjanovic et al., 2024).

Prototype-based interpretable methods have shown the value of visual evidence from historical cases (Kim et al., 2021; Wang et al., 2025). Nevertheless, their applicability to clinically oriented disease screening remains limited. Existing methods are typically designed for diagnosis or multi-label recognition, where predictions are explained by a fixed set of learned prototypes associated with predefined disease classes. In contrast, clinically oriented disease screening requires maintaining scalable evidence repositories of both normal and pathological historical cases, retrieving query-specific region-level evidence, and integrating such evidence into the prediction process. How to achieve these capabilities remains underexplored.

To address these gaps, we introduce **EviScreen, an evidential reasoning** framework for disease screening with medical images (Figure 1c). Our framework provides **retrospection interpretability** with similar image patches from dual knowledge banks of normal and pathological historical cases, making its reasoning process more transparent and reliable. Unlike previous methods that rely on saliency maps or deviation maps, `EviScreen` advances **localization interpretability** with abnormality maps generated by contrastive retrieval from dual knowledge banks.

## 1.1. Current Limitations

We identify three primary limitations in developing interpretable real-world disease screening methods:

**Lack of a clinically oriented evaluation framework.** While disease screening is often framed as an anomaly detection problem, existing evaluation protocols fall short of real-world clinical needs. Firstly, standard performance metrics, such as the area under the ROC curve (AUROC), are not aligned with clinical requirements. Secondly, current benchmarks often fail to assess model generalizability, as they typically lack real-world testing on external test sets.

**Limited evidence awareness in mainstream screening pipelines.** The interpretability of many models relies on post-hoc saliency maps. While these maps highlight regions deemed important, they lack case-based evidence explaining why a region appears pathological, limiting their alignment with expert reasoning. Prototype-based interpretable methods partially address this issue by comparing image regions with learned prototypes or historical examples. However, their representational capacity is bounded by a fixed number of prototypes specified before training, which may be insufficient to cover the diverse pathological appearances encountered in real-world screening.

**Insufficient mining of granular evidence from pathological cases.** Deviation-based anomaly detection methods can effectively identify what deviates from normal, but cannot leverage the rich, granular information contained within pathological cases. The core challenge lies in the fact that a pathological image comprises *both normal and pathological regions*, which makes it difficult to directly model the "pathological" class at the patch level.

## 1.2. Contributions

This paper addresses the aforementioned limitations by making the following key contributions:

**1. We propose a new, clinically relevant evaluation framework** for real-world disease screening. It uses ten public datasets across three critical medical modalities, prioritizing the proposed clinically oriented metrics and tests on external datasets. This setting mirrors real-world clinical scenarios, providing convincing benchmarks.

**2. We introduce `EviScreen`, an evidential reasoning framework** powered by the dual knowledge banks to facilitate real-world disease screening. The framework enhances prediction transparency through a synergy of retrospection interpretability and localization interpretability.

**3. The proposed dual knowledge banks** provide a scalable alternative to fixed prototype-based representations, capturing a broader spectrum of regional features from both normal and pathological cases. This enables our model to learn a rich representation of both normal regions and diverse pathological patterns, facilitating a more precise evidence-aware reasoning process.

**4. Comprehensive experiments** validate that the proposed `EviScreen` outperforms different types of comparative

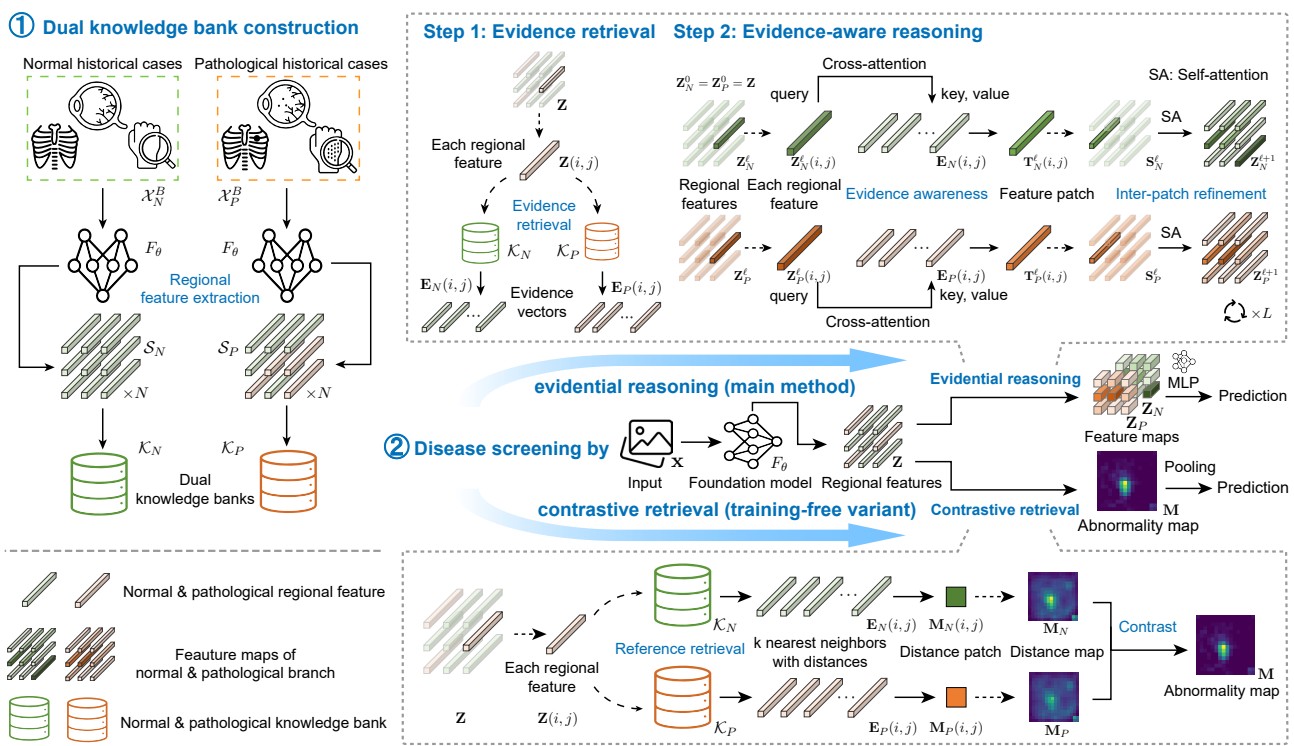

*Figure 2.* Overview of our framework that consists of two main stages: **(1) Dual knowledge bank construction**, where patch-level features from historical normal and pathological cases are extracted by a foundation model to construct two distinct knowledge banks, $\mathcal{K}_N$ and $\mathcal{K}_P$. **(2) Disease screening by evidential reasoning (main method) or contrastive retrieval (training-free variant)**. • *Evidential reasoning* includes two steps, the first step is evidence retrieval, where regional features from the current image input $\mathbf{x}$ are extracted to serve as queries. These queries retrieve the $k$-nearest neighbors from both knowledge banks, which serve as the evidence for subsequent reasoning ($\mathbf{E}_N$, $\mathbf{E}_P$). During the second step of evidence-aware reasoning, evidence awareness is realized by cross-attention between the current case and retrieved evidence, followed by inter-patch refinement via self-attention. • During the training-free variant with ***contrastive retrieval***, distance maps ($\mathbf{M}_N$, $\mathbf{M}_P$) are generated based on the distance to $k$-nearest neighbors retrieved from the dual knowledge banks. The final abnormality map $\mathbf{M}$ is obtained by contrasting the dual distance maps.

methods for real-world disease screening, particularly with respect to the clinically oriented metrics.

## 2. Related work

### 2.1. Interpretability in AI for Medical Imaging

Interpretability is critical for building trust in medical imaging applications. Conventional models that rely on post-hoc saliency maps (Selvaraju et al., 2017; Chattopadhay et al., 2018) to highlight regions of interest are often considered to provide interpretations of unsatisfactory quality (Saporta et al., 2022). Several newer methods move beyond mere feature attribution by employing techniques such as counterfactual intervention (Pan et al., 2025), graph networks (Hu et al., 2024), and natural language descriptions (Cai et al., 2024a). Nevertheless, these methods still do not provide a rationale by referring to historical cases in the way human experts do. While prototype-based interpretable models (Kim et al., 2021; Wang et al., 2025) improve transparency by associating predictions with representative visual prototypes, their

fixed-prototype design can limit capacity when real-world screening involves highly diverse appearances.

### 2.2. Disease Detection by Medical Anomaly Detection

Medical anomaly detection aims to distinguish abnormal cases from normal ones based on their deviation from normality (Cai et al., 2025; Li et al., 2025; Guo et al., 2023; Fernando et al., 2021). BMAD established benchmarks for medical anomaly detection, but it was not designed for clinically oriented disease screening (Bao et al., 2024). BenchReAD focused on retinal imaging to enhance the systematicity for medical anomaly detection (Lian et al., 2025). In this paper, we regard medical anomaly detection as one category of deviation-based prediction methods.

### 2.3. Coreset-Based Memory Bank

Coresets have been widely used for $k$-NN and $k$-Means approaches (Har-Peled & Kushal, 2005), representation learning (Roth et al., 2020), and deviation-based anomaly detec-

tion methods (Roth et al., 2022; Jiang et al., 2022). Typically, a coreset-based memory bank is constructed by extracting features from intermediate layers of ImageNet-pretrained models (Deng et al., 2009) and subsampling to reduce redundancy. In this paper, we construct dual knowledge banks using coreset-based memory banks to store regional features from both normal and pathological cases.

## 2.4. Vision Foundation Models for Medical Imaging

Vision foundation models, including general ones (Oquab et al., 2024; He et al., 2022) and those tailored for medical images (Zhou et al., 2023b;a; Yan et al., 2025; Yang et al., 2025), have shown promising performance on medical imaging (Zhang & Metaxas, 2024; Moor et al., 2023; Chen et al., 2022; Zhang et al., 2024). However, the adoption of foundation models for disease screening remains understudied. In this work, we adopt vision foundation models to extract regional features for dual knowledge bank construction.

## 3. Method

**Overview.** As illustrated in Figure 2, our proposed framework comprises two primary stages: *dual knowledge bank construction* (Section 3.1) and *evidential reasoning* (Section 3.2). In the first stage, we extract intermediate regional features from historical normal and pathological cases using a pretrained foundation model. These features are subsequently subsampled to form compact knowledge banks that represent normal and pathological patterns. In the second stage, *evidence is retrieved* from the dual knowledge banks to enable interpretable disease screening (Section 3.2.1). *Evidence-aware reasoning* is performed by first using cross-attention for evidence awareness, followed by self-attention for inter-patch refinement (Section 3.2.2). Additionally, our framework enhances training-free disease screening through the proposed *contrastive retrieval* (Section 3.3).

## 3.1. Dual Knowledge Bank Construction

Let $\mathcal{X}_N$ and $\mathcal{X}_P$ denote the training sets of normal and pathological cases, respectively, where $\forall x \in \mathcal{X}_N : y_x = 0$ and $\forall x \in \mathcal{X}_P : y_x = 1$. We partition each set into two disjoint subsets: one for constructing the dual knowledge banks ($\mathcal{X}_N^B, \mathcal{X}_P^B$) and the other for training the evidential reasoning module ($\mathcal{X}_N^R, \mathcal{X}_P^R$), i.e., $\mathcal{X}_N = \mathcal{X}_N^B \cup \mathcal{X}_N^R$ and $\mathcal{X}_P = \mathcal{X}_P^B \cup \mathcal{X}_P^R$. Using a frozen foundation model $F_\theta$, we extract intermediate regional features from all images in $\mathcal{X}_N^B$ and $\mathcal{X}_P^B$, yielding two sets of regional features:

$$\mathcal{S}_N = \bigcup_{x_i \in \mathcal{X}_N^B} \mathcal{G}_{agg}\left(F_\theta\left(x_i\right)\right), \tag{1a}$$

$$\mathcal{S}_P = \bigcup_{x_i \in \mathcal{X}_P^B} \mathcal{G}_{agg}\left(F_\theta\left(x_i\right)\right), \tag{1b}$$

where $\mathcal{G}_{agg}$ represents a locally aware patch-feature aggregation function (Roth et al., 2022). In addition, to reduce redundancy and enhance efficiency, we apply greedy coreset subsampling (Agarwal et al., 2005; Roth et al., 2022) to $\mathcal{S}_N$ and $\mathcal{S}_P$, producing compact knowledge banks $\mathcal{K}_N$ and $\mathcal{K}_P$. The optimization objective is to find a subset that best represents the entire feature set:

$$\mathcal{K}_N^* = \arg \min_{\mathcal{K}_N \subset \mathcal{S}_N} \max_{m \in \mathcal{S}_N} \min_{n \in \mathcal{K}_N} \|m - n\|_2, \tag{2a}$$

$$\mathcal{K}_P^* = \arg \min_{\mathcal{K}_P \subset \mathcal{S}_P} \max_{m \in \mathcal{S}_P} \min_{n \in \mathcal{K}_P} \|m - n\|_2, \tag{2b}$$

where $\| \cdot \|_2$ denotes the Euclidean distance. Since this problem is NP-hard, an iterative greedy approximation is employed (Sener & Savarese, 2018; Wolsey & Nemhauser, 2014; Roth et al., 2022). The resulting dual knowledge banks provide the foundational evidence for the subsequent reasoning stage. In addition to their role as evidence providers in the reasoning module, the banks themselves enhance training-free disease screening through the proposed contrastive retrieval, which we discuss in Section 3.3.

## 3.2. Disease Screening by Evidential Reasoning

Given an input image $\mathbf{x} \in \mathbb{R}^{H \times W \times C}$, we first extract its intermediate regional features using the same frozen foundation model $F_\theta$ and aggregation function $\mathcal{G}_{agg}$:

$$\mathbf{Z} = \mathcal{G}_{agg}(F_\theta\left(\mathbf{x}\right)), \tag{3}$$

where $\mathbf{Z} \in \mathbb{R}^{h \times w \times d}$ is the feature map with spatial dimensions $h \times w$ and feature dimension $d$. Each regional feature vector $\mathbf{Z}\left(i, j\right) \in \mathbb{R}^d$ in $\mathbf{Z}$ serves as a query.

### 3.2.1. EVIDENCE RETRIEVAL

For each regional feature vector $\mathbf{Z}\left(i, j\right)$ (query), we retrieve the $k$-nearest neighbors from both the normal knowledge bank $\mathcal{K}_N$ and the pathological knowledge bank $\mathcal{K}_P$. This process yields two sets of evidence vectors $\mathbf{E}_N\left(i, j\right)$ and $\mathbf{E}_P\left(i, j\right)$ for each spatial location $(i, j)$:

$$\mathbf{E}_N\left(i, j\right) = \mathrm{NN}(\mathbf{Z}\left(i, j\right), k; \mathcal{K}_N), \forall i \in \{1, \cdots, h\},$$
$$j \in \{1, \cdots, w\}, \tag{4a}$$

$$\mathbf{E}_P\left(i, j\right) = \mathrm{NN}(\mathbf{Z}\left(i, j\right), k; \mathcal{K}_P), \forall i \in \{1, \cdots, h\},$$
$$j \in \{1, \cdots, w\}, \tag{4b}$$

where the function $\mathrm{NN}\left(\mathbf{z}, k; \mathcal{K}\right)$ returns a set of $k$ closest vectors in knowledge bank $\mathcal{K}$ along with the Euclidean distances to them. Therefore, $\mathbf{E}_N, \mathbf{E}_P \in \mathbb{R}^{h \times w \times k \times d}$.

### 3.2.2. EVIDENCE-AWARE REASONING

The retrieved evidence above is then used to conduct evidence-aware reasoning and generate evidence-aware feature maps $\mathbf{Z}_N$ and $\mathbf{Z}_P$. We denote the initial query as

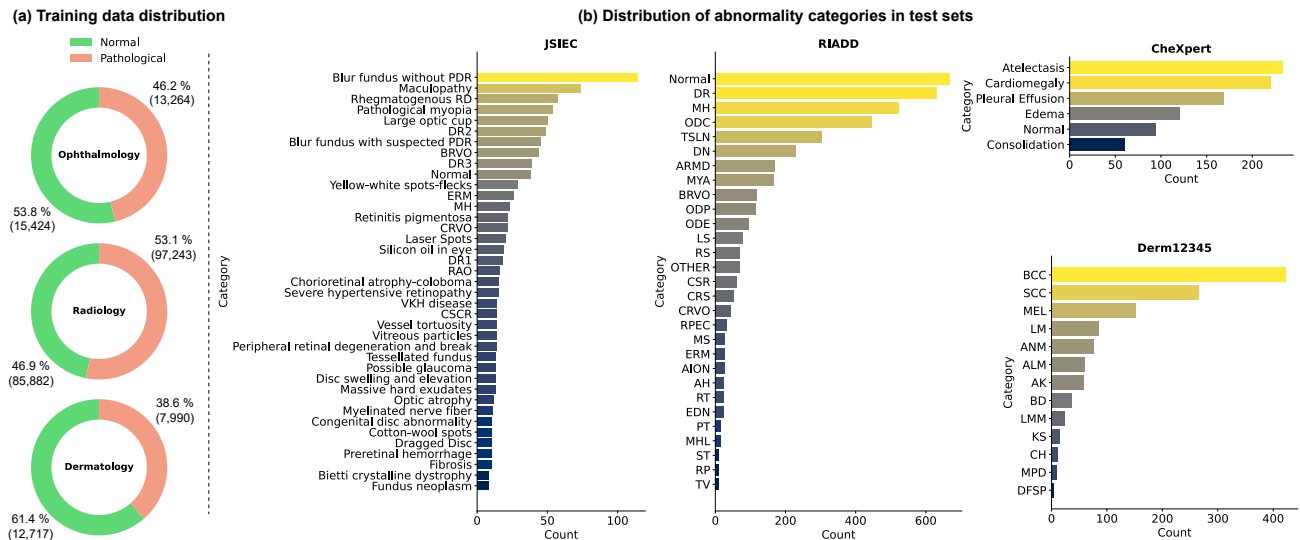

*Figure 3.* (a) Training data distribution. (b) Distribution of abnormality categories in test sets (JSIEC, RIADD, CheXpert, and Derm12345).

$\mathbf{Z}_N^0(i,j) = \mathbf{Z}(i,j)$. For each layer $l = 0, \ldots, L-1$, we first perform evidence-aware cross-attention. The evidence vectors from $\mathbf{E}_N(i,j)$ and $\mathbf{E}_P(i,j)$ are linearly projected to serve as keys and values. For the branch with the normal knowledge bank (similarly for the pathological branch):

$$\mathbf{T}_N^\ell(i,j) = \text{softmax}\left(\frac{\mathbf{Z}_N^\ell(i,j)(\mathbf{E}_N(i,j))^\top}{\sqrt{d}}\right)\mathbf{E}_N(i,j),$$
$$\forall i \in \{1, \cdots, h\}, j \in \{1, \cdots, w\}. \qquad (5)$$

To achieve inter-patch refinement, we reshape $\mathbf{T}_N^\ell(i,j) \in \mathbb{R}^{h \times w \times d}$ to $\mathbf{S}_N^\ell(i,j) \in \mathbb{R}^{(h \cdot w) \times d}$ and apply self-attention to $\mathbf{S}_N^\ell(i,j)$, generating the feature map of the next layer:

$$\mathbf{Z}_N^{l+1} = \text{softmax}\left(\frac{\mathbf{S}_N^\ell(\mathbf{S}_N^\ell)^\top}{\sqrt{d}}\right)\mathbf{S}_N^\ell. \qquad (6)$$

Finally, we obtain evidence-aware feature map $\mathbf{Z}_N = \mathbf{Z}_N^{L-1}$, and similarly, $\mathbf{Z}_P = \mathbf{Z}_P^{L-1}$. The prediction is obtained from the aggregated evidence-aware features through a multi-layer perceptron (MLP):

$$\hat{y} = \text{MLP}\left([\mathbf{Z}_N^{\text{CLS}}; \mathbf{Z}_P^{\text{CLS}}]\right), \qquad (7)$$

where $\mathbf{Z}^{\text{CLS}}$ refers to the [CLS] token of $\mathbf{Z}$.

### 3.3. Training-Free Variant with Contrastive Retrieval

The proposed dual knowledge banks not only provide the evidence for evidential reasoning but can also enhance training-free disease screening. Leveraging the pathological knowledge bank is nontrivial since the regional features extracted from pathological cases are not "clean" (containing both normal and pathological image regions). To address this challenge, we propose contrastive retrieval, which contrasts

the distance maps produced by retrieving from the dual knowledge banks to generate the abnormality map.

**Contrastive retrieval.** Following Equation Equation (3), we obtain the feature map $\mathbf{Z} \in \mathbb{R}^{h \times w \times d}$ of the input image $\mathbf{x} \in \mathbb{R}^{H \times W \times C}$. For each regional feature vector $\mathbf{Z}(i,j)$, we calculate the average distance to the top $k$-nearest neighbors from both the normal knowledge bank $\mathcal{K}_N$ and the pathological knowledge bank $\mathcal{K}_P$. This process yields two distance maps, where for each spatial location $(i,j)$:

$$\mathbf{M}_N(i,j) = \text{NN-Dis}\left(\mathbf{Z}(i,j), k; \mathcal{K}_N\right),$$
$$\forall i \in \{1, \cdots, h\}, j \in \{1, \cdots, w\}, \qquad (8a)$$
$$\mathbf{M}_P(i,j) = \text{NN-Dis}\left(\mathbf{Z}(i,j), k; \mathcal{K}_P\right),$$
$$\forall i \in \{1, \cdots, h\}, j \in \{1, \cdots, w\}, \qquad (8b)$$

where $\mathbf{M}_N, \mathbf{M}_P \in \mathbb{R}^{h \times w}$, and NN-Dis$(\mathbf{z}, k; \mathcal{K})$ outputs the average Euclidean distances to the nearest $k$ vectors in the knowledge bank $\mathcal{K}$. We generate the abnormality map by contrasting the two distance maps:

$$\mathbf{M}(i,j) = \text{ReLU}\left(\mathbf{M}_N(i,j) - \mathbf{M}_P(i,j)\right),$$
$$\forall i \in \{1, \cdots, h\}, j \in \{1, \cdots, w\}. \qquad (9)$$

The final prediction score is calculated by pooling the point-wise scores of the abnormality map $\mathbf{M}$.

## 4. Experiments and Results

### 4.1. Evaluation Framework Construction

To the best of our knowledge, there is no clinically oriented evaluation framework for real-world disease screening. We construct the evaluation framework to facilitate this and future research, focusing on clinically oriented metrics and external tests that simulate clinical scenarios.

*Table 1.* Results for disease screening in four testing sets (%). The best results are in **bold**, and the second-best results are underlined.
* A variant of PatchCore that uses the same foundation models as ours.

| Dataset | Metric | Ours | Ours-TF | FM | PatchCore* | PatchCore | NFM-DRA | DRA | SCRD4AD | EDC | SimpleNet | CIPL |
|---|---|---|---|---|---|---|---|---|---|---|---|---|
| JSIEC | AUROC | **98.06** | 96.76 | 95.84 | 94.96 | 92.12 | 95.53 | 92.53 | 94.88 | 79.12 | 73.73 | 94.83 |
| | AP | **96.10** | 94.20 | 94.24 | 89.61 | 86.62 | 93.23 | 89.53 | 89.85 | 71.44 | 57.66 | 91.36 |
| | Spe@95%R | **94.74** | 91.48 | 87.95 | 87.26 | 81.09 | 90.37 | 80.12 | 88.50 | 51.45 | 53.81 | 87.33 |
| | Spe@99%R | **91.62** | 87.74 | 79.29 | 83.31 | 74.31 | 84.07 | 71.88 | 78.74 | 43.01 | 49.31 | 79.57 |
| | Spe@100%R | **91.27** | 87.33 | 78.39 | 82.34 | 73.75 | 82.96 | 71.12 | 78.46 | 42.11 | 49.10 | 78.60 |
| | CSR | **88.95** | 85.07 | 80.02 | 68.70 | 68.67 | 81.61 | 69.81 | 70.72 | 37.16 | 38.27 | 73.41 |
| RIADD | AUROC | **91.32** | 90.42 | 87.88 | 87.49 | 79.56 | 84.63 | 81.62 | 83.83 | 57.23 | 60.21 | 87.09 |
| | AP | **60.35** | 59.96 | 38.16 | 59.73 | 32.46 | 40.97 | 50.45 | 39.94 | 11.67 | 13.44 | 59.08 |
| | Spe@95%R | **72.92** | 69.96 | 65.28 | 61.64 | 48.70 | 56.50 | 51.62 | 57.92 | 16.44 | 24.58 | 64.92 |
| | Spe@99%R | **59.39** | 55.26 | 55.73 | 48.23 | 33.62 | 42.68 | 36.71 | 47.73 | 9.51 | 15.86 | 51.40 |
| | Spe@100%R | **55.35** | 51.25 | 51.75 | 41.63 | 29.36 | 37.64 | 29.74 | 43.98 | 9.08 | 14.26 | 43.99 |
| | CSR | **54.38** | 50.73 | 48.91 | 42.36 | 28.25 | 36.27 | 30.38 | 42.64 | 8.94 | 13.94 | 44.85 |
| CheXpert | AUROC | **96.72** | 92.30 | 95.60 | 67.41 | 57.92 | 63.94 | 85.63 | 51.50 | 65.26 | 55.24 | 92.57 |
| | AP | **94.71** | 82.22 | 91.81 | 38.15 | 30.61 | 34.23 | 70.64 | 27.74 | 33.88 | 29.90 | 84.60 |
| | Spe@95%R | **84.04** | 66.60 | 79.79 | 23.19 | 22.13 | 25.11 | 45.96 | 12.55 | 25.11 | 11.49 | 68.09 |
| | Spe@99%R | **74.26** | 60.00 | 67.66 | 16.81 | 11.49 | 16.17 | 23.62 | 5.53 | 16.81 | 9.15 | 48.72 |
| | Spe@100%R | **68.72** | 55.74 | 60.00 | 15.53 | 11.06 | 13.62 | 18.72 | 5.53 | 14.89 | 8.94 | 49.15 |
| | CSR | **70.59** | 48.91 | 57.41 | 12.61 | 8.57 | 10.54 | 17.63 | 4.61 | 12.52 | 7.58 | 47.29 |
| Derm12345 | AUROC | **97.43** | 97.21 | 95.93 | 84.10 | 81.59 | 85.27 | 94.97 | 74.30 | 75.71 | 75.33 | 94.00 |
| | AP | 34.23 | **37.23** | 28.41 | 3.24 | 4.49 | 5.69 | 19.70 | 3.80 | 2.79 | 4.22 | 13.37 |
| | Spe@95%R | **90.29** | 88.27 | 84.37 | 57.24 | 51.56 | 57.85 | 77.19 | 45.73 | 36.49 | 38.10 | 82.04 |
| | Spe@99%R | **85.78** | 80.27 | 73.57 | 50.29 | 43.50 | 49.21 | 66.16 | 38.03 | 25.10 | 25.95 | 68.08 |
| | Spe@100%R | **78.49** | 70.35 | 55.41 | 46.90 | 36.82 | 41.80 | 56.91 | 32.86 | 22.15 | 20.94 | 54.95 |
| | CSR | **77.92** | 69.94 | 55.09 | 46.62 | 36.67 | 41.63 | 56.66 | 32.69 | 22.04 | 20.84 | 54.74 |

### 4.1.1. BENCHMARKS AND DATA

Benchmarks are established across three medical domains (ophthalmology, radiology, and dermatology) using ten public datasets spanning three modalities. The datasets include color fundus photography (JSIEC (Cen et al., 2021), RIADD (Pachade et al., 2021), EDDFS (Xia et al., 2024), BRSET (Nakayama et al., 2024)); chest X-rays (CheXpert (Irvin et al., 2019), MIMIC-CXR (Johnson et al., 2019a)); and dermoscopic images (Derm12345 (Yilmaz et al., 2024), HAM10000 (Tschandl et al., 2018), BCN20000 (Hernández-Pérez et al., 2024), PAD-UFES-20 (Pacheco et al., 2020)). Figure 3 illustrates the distribution of training data and abnormality categories in test sets. Refer to Section A of Appendix for more details.

### 4.1.2. CLINICALLY ORIENTED METRICS PROPOSED

**Specificity at X% Recall.** In clinical practice, disease screening demands high recall to minimize missed diagnoses. Concurrently, maximizing specificity is a key objective to reduce unnecessary follow-up tests. Based on this clinical need, we introduce the metric Specificity at X% Recall ($\mathrm{Spe}@X\%\mathrm{R}$), formally calculated as follows:

First, we define the set of decision thresholds, $\mathcal{T}_{\geq X\%}$, to include all thresholds $\tau$ for which the recall is at least $X\%$:

$$\mathcal{T}_{\geq X\%} = \{\tau \mid \mathrm{Recall}(\tau) \geq X/100\} . \quad (10)$$

The value of $\mathrm{Spe}@X\%\mathrm{R}$ is defined as the maximum specificity achieved by any threshold in the set $\mathcal{T}_{\geq X\%}$:

$$\mathrm{Spe}@X\%\mathrm{R} = \max_{\tau \in \mathcal{T}_{\geq X\%}} \{\mathrm{Specificity}(\tau)\} . \quad (11)$$

**Clear Separation Rate.** Another key goal for AI in disease screening is to reduce the manual review workload for clinicians to double-check ambiguous cases. To this end, we introduce a metric termed Clear Separation Rate (CSR), which quantifies the proportion of cases that fall outside the overlapping prediction score region between the two classes, i.e., all negative cases score below any positive case and vice versa. Let $s_i$ be the predicted score for the $i$-th case and $y \in \{0, 1\}$ be its true label, CSR is calculated as:

$$\mathrm{CSR} = \frac{1}{N} \left( \sum_{i=1}^{N} \mathbb{I}(s_i < \min_{j:y_j=1} s_j) \right.$$
$$\left. + \sum_{i=1}^{N} \mathbb{I}(s_i > \max_{j:y_j=0} s_j) \right), \quad (12)$$

where $N$ is the sample size and $\mathbb{I}(\cdot)$ is the indicator function.

In our experiments, we conduct a comprehensive evaluation using multiple metrics, including the average AUROC, Average Precision (AP), Spe@95%R, Spe@99%R, Spe@100%R, and CSR.

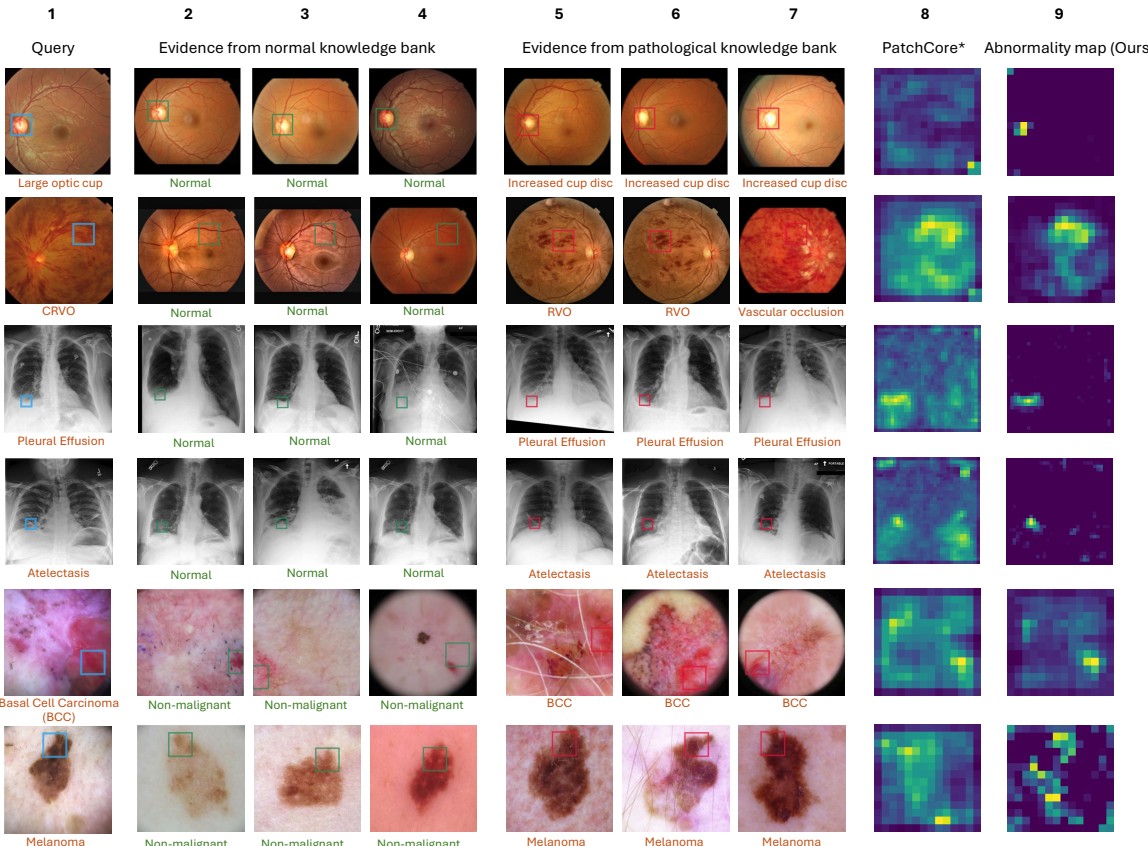

*Figure 4.* Visualization showing our method provides interpretable evidence when making predictions. Columns 2-7 depict the regional evidence for each representative query patch retrieved from historical cases, providing **retrospection interpretation**. In addition, the last column represents abnormality maps, providing **localization interpretation**. Specifically, the first column includes cases for testing, with representative query patches highlighted in blue squares. The 2-4 columns depict the retrieved evidence from the normal knowledge bank (patches in green squares), and the whole source images for the patches are shown for reference. Similarly, the 5-7 columns depict the retrieved evidence from the pathological knowledge bank (patches in red squares). The final two columns compare the localization interpretability, showing our method provides more focused abnormality maps than PatchCore while using the same foundation models.

## 4.2. Results on Real-World Disease Screening

**EviScreen outperforms various types of comparative approaches for real-world disease screening.** We evaluate `EviScreen` on our proposed evaluation framework across three clinical domains, comparing against various types of state-of-the-art methods. The compared methods include direct prediction by fine-tuning foundation models (FM), deviation-based anomaly detection approaches (PatchCore (Roth et al., 2022), SCRD4AD (Li et al., 2025), EDC (Guo et al., 2023), and SimpleNet (Liu et al., 2023)), supervised variants (NFM-DRA (Lian et al., 2025), DRA (Ding et al., 2022)), and a prototype-based interpretable method (CIPL (Wang et al., 2025)). As shown in Table 1, `EviScreen` outperforms various approaches for real-world disease screening, consistently achieving the highest AUROC, AP, Spe@$X$%R, and CSR, across all datasets. Notably, the consistent gains of `EviScreen` over the recent prototype-based CIPL model suggest that scal-

able dual knowledge banks are better suited to real-world screening than fixed learned prototypes. To provide a more comprehensive evaluation, we also include a training-free variant of our approach (Ours-TF). Further discussions are provided in Section 4.3.

**EviScreen exhibits a notable improvement with regard to clinically oriented metrics.** AUROC is widely used in binary classification and anomaly detection to evaluate the ability of models to distinguish between positive and negative cases. However, we observe that models with small differences in AUROC may exhibit significant differences in clinically oriented metrics, as shown in Table 1.

In terms of the Spe@$X$%R metric, `EviScreen` outperforms other approaches by notable margins, even when improvements in AUROC are relatively modest. For instance, on the JSIEC dataset, `EviScreen` produces relative improvements over FM of 2.3% in AUROC, but by

*Table 2.* Ablation analysis results (%) on components of evidential reasoning: evidence retrieval and evidence-aware reasoning.

| Evidence retrieval | Evidence-aware reasoning | JSIEC | | | | | RIADD | | | | |
|---|---|---|---|---|---|---|---|---|---|---|---|
| | | AUROC | Spe@95R | Spe@99R | Spe@100R | CSR | AUROC | Spe@95R | Spe@99R | Spe@100R | CSR |
| ✓ | | 96.76 | 91.48 | 87.74 | 87.33 | 85.07 | 90.42 | 69.96 | 55.26 | 51.25 | 50.73 |
| | ✓ | 97.78 | 92.73 | 88.02 | 86.22 | 86.99 | 86.12 | 61.86 | 47.97 | 41.93 | 41.53 |
| ✓ | ✓ | **98.06** | **94.74** | **91.62** | **91.27** | **88.95** | **91.32** | **72.92** | **59.39** | **55.35** | **54.38** |

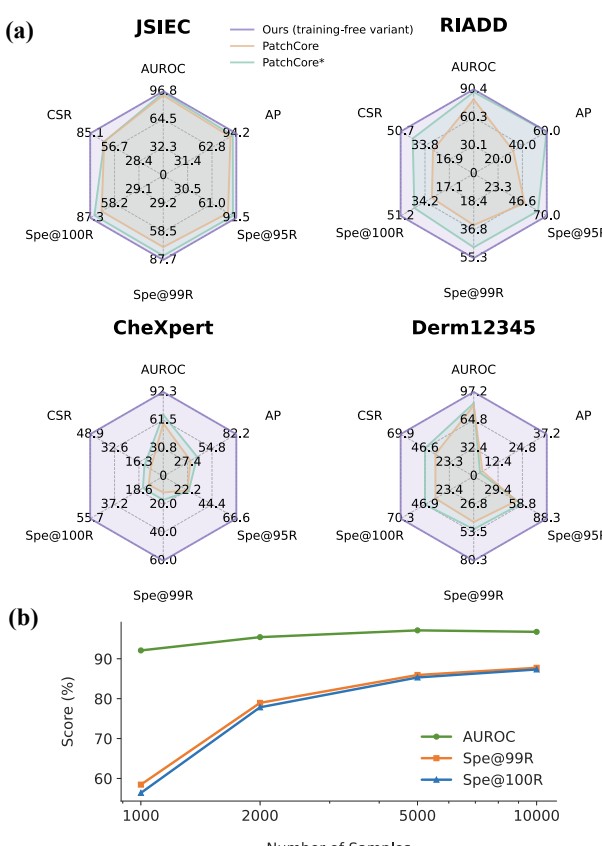

*Figure 5.* (a) Results (%) of ours (training-free variant) compared to other training-free methods. (b) Performance improves when the number of samples increases, especially for Spe@$X$%R.

7.7%, 15.6%, and 16.43% in Spe@95%R, Spe@99%R, and Spe@100%R, respectively. These advantages could meaningfully reduce clinical costs by minimizing the need for re-examinations.

Similar trends are observed for the metric of CSR that measures the separation between predictions of negative and positive cases, where `EviScreen` surpasses the best runner-up (excluding Ours-TF) by 9.0%, 11.2%, 23.0%, and 37.5%, on the four datasets, respectively. These results show that `EviScreen` achieves clearer separation between negative and positive cases, indicating its potential to reduce the workload of clinicians in double-checking AI predictions.

**`EviScreen` provides interpretable evidence when mak-**

**ing predictions.** As shown in Figure 4, the interpretability of `EviScreen` is two-fold. Columns 2-7 depict **retrospection interpretability** by tracing historical cases, generating regional evidence for each representative query patch from both the normal and pathological knowledge bank. Column 9 illustrates the **localization interpretability** presented by abnormality maps. By comparing our abnormality maps and the deviation maps produced by another training-free method (PatchCore*), it can be observed that our method generates more focused maps, providing clearer interpretation. More visualizations are in Section C of Appendix.

### 4.3. Dual Knowledge Banks Enhance Training-Free Disease Screening via Contrastive Retrieval

Beyond providing evidence for evidential reasoning, the dual knowledge banks have the potential to enable training-free disease screening. To achieve this, we propose contrastive retrieval to capture the discrepancies between the retrieved neighbors from the dual knowledge banks. The pipeline of the training-free variant of our method is illustrated in Figure 2 (refer to Section 3.3 for more details).

**Performance compared to other training-free methods.** Figure 5a highlights the advantage of our dual knowledge banks with contrastive retrieval over previous training-free methods. As illustrated by the radar charts, the foundation model-enhanced PatchCore (PatchCore*) outperforms the original version using ImageNet-pretrained feature extractors. Crucially, even when adopting the same foundation models, our method consistently outperforms PatchCore across all benchmarks and various evaluation metrics.

**Scaling the dual knowledge banks.** As Figure 5b shows, the performance of our training-free methods with the dual knowledge banks improves as the number of samples increases, in particular for clinically oriented metrics.

### 4.4. Ablation Study

**Ablation analysis on evidential reasoning.** The proposed evidential reasoning consists of two key components: evidence retrieval and evidence-aware reasoning. The former supplies query-specific visual evidence from dual knowledge banks of normal and pathological cases, whereas the latter enables the model to incorporate such evidence into the prediction process. As shown in Table 2, performance

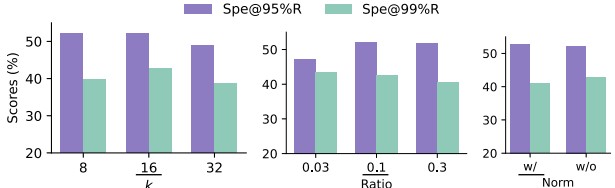

*Figure 6.* Hyperparameter analysis on the number of nearest neighbors $k$, ratio of subsampling, and normalization before calculating Euclidean distances. Default options are underlined.

decreases when any of the components is absent, showing the necessity of these mechanisms.

**Hyperparameter analysis on dual knowledge bank construction.** We conduct hyperparameter analysis on the number of nearest neighbors $k$ (Section 3.2.1), ratio of subsampling (Section 3.1), and the normalization before calculating Euclidean distance. Figure 6 illustrates the performance in the validation set of the ophthalmology benchmark, using the dual knowledge banks with contrastive retrieval. Hyperparameters selected in our default settings are underlined.

### 4.5. Implementation Details

Our code is implemented using PyTorch 2.4.1 (Paszke et al., 2019). All experiments are carried out with Nvidia GeForce RTX 3090 GPUs. We employ state-of-the-art ViT-based (Dosovitskiy et al., 2020) foundation models for each modality: RETFound-Dinov2 (Zhou et al., 2025) for color fundus photography (CFP), CheXFound (Yang et al., 2025) for chest X-rays, and PanDerm (Yan et al., 2025) for dermoscopic images. Faiss 1.8.0 (Johnson et al., 2019b) serves as the engine for dual knowledge banks. For evidential reasoning, AdamW (Loshchilov & Hutter, 2017) is adopted as the default optimizer, with a weight decay of 0.05, $\beta_1$ of 0.9, and $\beta_2$ of 0.95. We employ a "warm-up" strategy by linearly increasing the learning rate (selected by the performance in the validation set from 1.25e-4, 2e-4, and 2.5e-4) to the desired value and then decreasing it using a cosine decay schedule. Batch size is 32 for CFP ($224 \times 224$) and dermoscopic images, and 8 ($512 \times 512$) for chest X-rays. Refer to Section B of Appendix for detailed descriptions.

## 5. Discussion and Conclusion

In this paper, we systematically address the challenge of interpretable real-world disease screening by introducing evidential reasoning to enhance performance and interpretability simultaneously. Specifically, we first establish a comprehensive evaluation framework featuring novel clinically oriented metrics and extensive datasets for three clinical domains. The proposed `EviScreen` provides both retrospection interpretability and localization interpretability alongside its predictions, showing higher performance and bet-

ter interpretability than current deviation-based prediction methods and direct prediction methods. Extensive experiments consistently validate the improvements our method brings, in particular for clinically oriented metrics. We hope that this work can inspire more clinically oriented, interpretable algorithms for real-world disease screening. Future work will address broader modalities, 3D medical images, and finer-grained screening tasks.

## Impact Statement

The paper presents work that aims to advance the field of medical image analysis by introducing an interpretable framework for disease screening, mimicking clinical decision-making through evidential reasoning. By prioritizing high specificity at clinical-level recall and providing transparent visual evidence from historical cases, our method has the potential to significantly reduce unnecessary follow-up examinations and the associated psychological and economic burdens on patients. While the evidence-based mechanism enhances trust between AI and clinicians, potential ethical considerations, such as strict privacy safeguards when storing patient data in real-world clinical deployments, must be addressed.

## Acknowledgments

This work was supported in part by a Shenzhen-Hong Kong-Macao Science and Technology Plan Project (Category C Project) under Shenzhen Municipal Science and Technology Innovation Commission (project no. SGDX20230821092 359002) and a project under Innovation and Technology Support Programme of Hong Kong Innovation and Technology Commission (project no. ITS/202/23). C. Lian is supported in part by the Research Institute for Smart Ageing of the Hong Kong Polytechnic University.

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

## A. More Details of Benchmarks and Data

To establish a comprehensive evaluation framework, we develop benchmarks across three critical medical domains: ophthalmology, radiology, and dermatology. Figure 3 in the main paper illustrates the distribution of abnormality categories and the composition of the training set. This section provides detailed specifications to facilitate understanding and reproducibility.

### A.1. Ophthalmology

Our ophthalmology benchmark utilizes color fundus photography (CFP) with training and validation sets derived from EDDFS (Xia et al., 2024) and BRSET (Nakayama et al., 2024). We categorize samples based on abnormality presence: cases without detected abnormalities are classified as normal, while those exhibiting any abnormality are classified as pathological. To simulate real-world disease screening scenarios, we employ two external datasets: JSIEC (Cen et al., 2021) and RIADD (Pachade et al., 2021) as test sets.

### A.2. Radiology

For the radiology benchmark using chest X-rays, we utilize MIMIC-CXR (Johnson et al., 2019a) for training and validation. Our preprocessing pipeline retains only samples with "AP" or "PA" view positions and excludes samples containing uncertain labels. Normal cases are defined as those annotated with "No Finding", while pathological cases include samples with any abnormal findings except "Support Devices". We use the official CheXpert (Irvin et al., 2019) evaluation set as our test dataset, chosen for its board-certified radiologist annotations ensuring label reliability. Following the original CheXpert recommendations, our evaluation focuses on five primary pathological categories: atelectasis, cardiomegaly, consolidation, edema, and pleural effusion.

### A.3. Dermatology

The dermatology benchmark incorporates dermoscopic images with training and validation sets constructed from HAM10000 (Tschandl et al., 2018), BCN20000 (Hernández-Pérez et al., 2024), and PAD-UFES-20 (Pacheco et al., 2020). Since dermoscopic examination inherently includes few completely normal samples because clinicians typically examine only areas that appear different from normal skin, we adapt our classification scheme accordingly. Samples annotated as "benign" are treated as "normal" cases, while "malignant" samples constitute the "pathological" cases for detection. We employ Derm12345 (Yilmaz et al., 2024) as our test set for evaluation.

## B. More Comprehensive Implementation Details

Our code is implemented using PyTorch 2.4.1 (Paszke et al., 2019). All experiments are carried out with Nvidia GeForce RTX 3090 GPUs. We employ state-of-the-art foundation models for each modality: RETFound-Dinov2 (Zhou et al., 2025) for CFP, CheXFound (Yang et al., 2025) for chest X-rays, and PanDerm (Yan et al., 2025) for dermoscopic images. The foundation models chosen as regional feature extractors are based on ViT-L (Dosovitskiy et al., 2020; Vaswani et al., 2017), which consists of 24 transformer blocks. Features in the layers of 7 and 17 are selected and aggregated by adaptive average pooling to generate desired regional features. RETFound-Dinov2 (Zhou et al., 2025) for CFP receives input resolution of 224×224 with the patch size of 14. CheXFound (Yang et al., 2025) for chest X-rays receives input resolution of 512×512 with the patch size of 16. PanDerm (Yan et al., 2025) for dermoscopic images receives input resolution of 224×224 with the patch size of 16. The dimension of evidence vectors is 1,024. Following previous related work (Roth et al., 2022; Lian et al., 2025), no data augmentation is applied to avoid including new abnormality or losing original abnormality in the images. During the construction and usage of dual knowledge banks, Faiss 1.8.0 (Johnson et al., 2019b) is adopted for the nearest neighbor retrieval and distance computations.

For evidential reasoning, AdamW (Loshchilov & Hutter, 2017) is adopted as the default optimizer, with a weight decay of 0.05, $\beta_1$ of 0.9, and $\beta_2$ of 0.95. We employ a "warm-up" strategy by linearly increasing the learning rate (selected based on validation performance from 1.25e-4, 2e-4, and 2.5e-4) to the desired value and then decreasing it using a cosine decay schedule. Batch size is 32 for CFP and dermoscopic images, and 8 for chest X-rays. The network for evidential reasoning is based on Transformers (Dosovitskiy et al., 2020; Vaswani et al., 2017), with an embedding dimension of 1,024, 8 attention heads, 256 patches, and a depth of 4 layers. Each transformer block comprises a cross-attention module for evidence awareness, followed by a self-attention module for inter-patch refinement. We adopt 2D unlearnable sin-cos positional

embeddings to introduce the positional information.

## C. More Examples of Interpretability Visualization

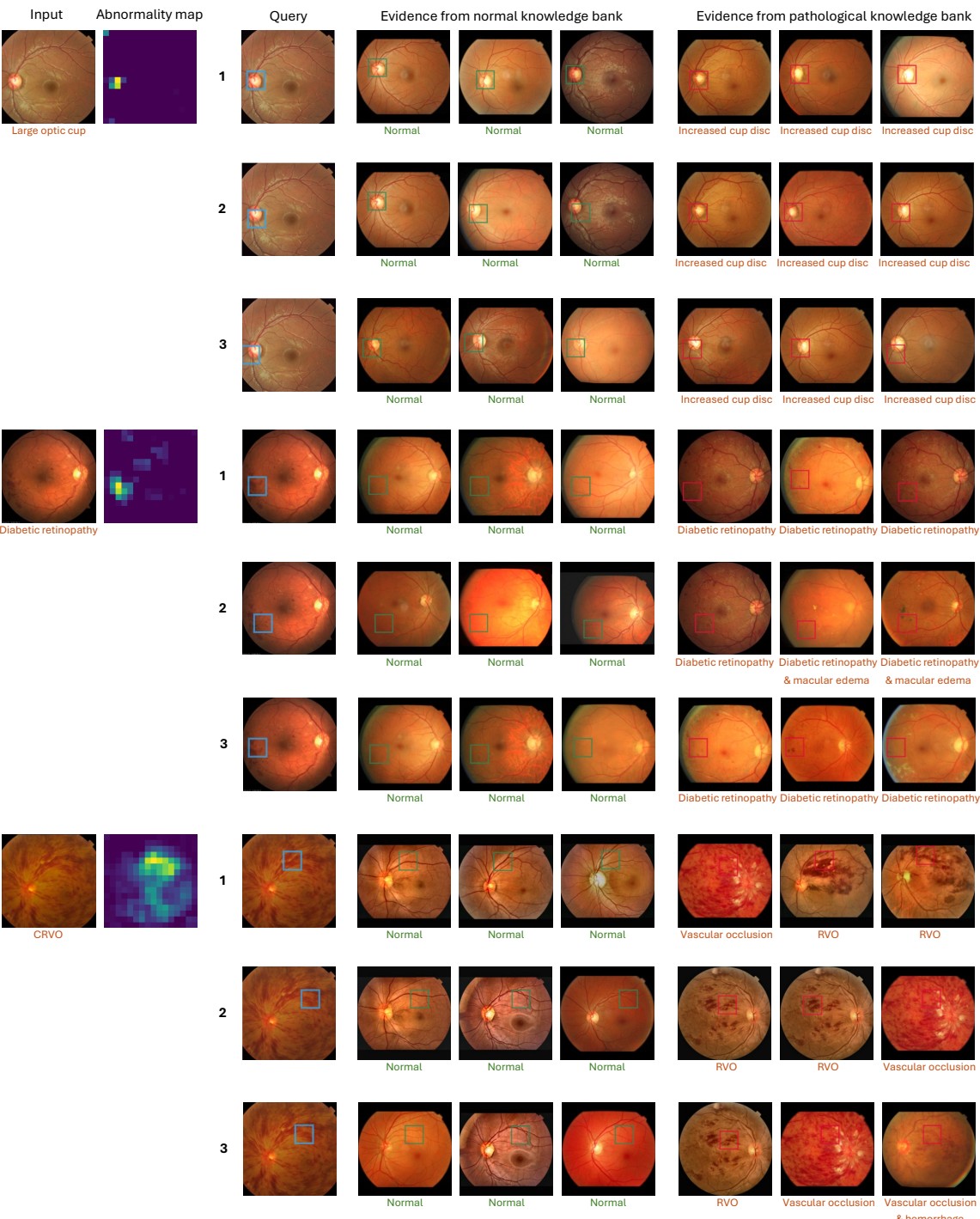

*Figure 7.* More examples of interpretability visualization for ophthalmology.

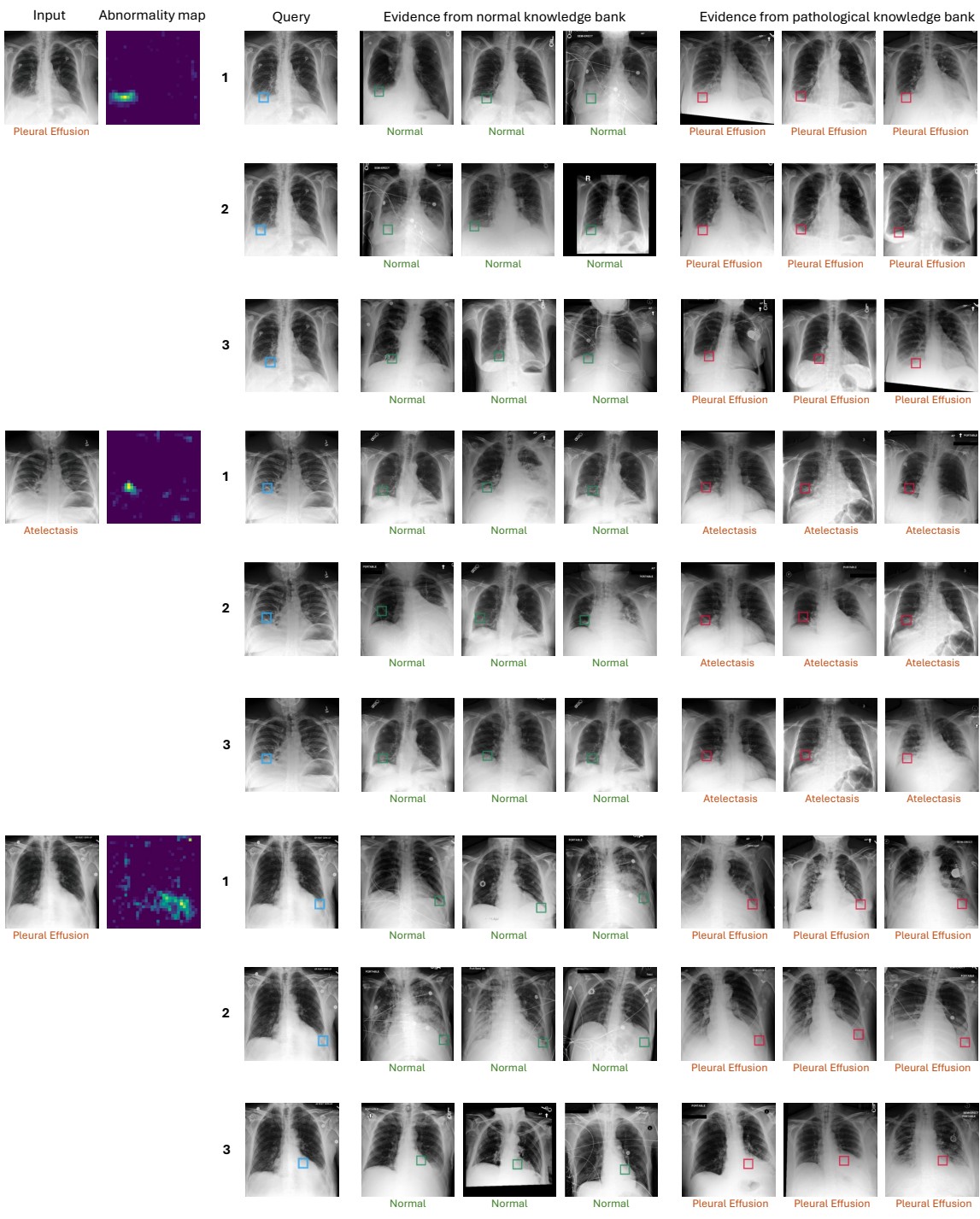

*Figure 8.* More examples of interpretability visualization for radiology.

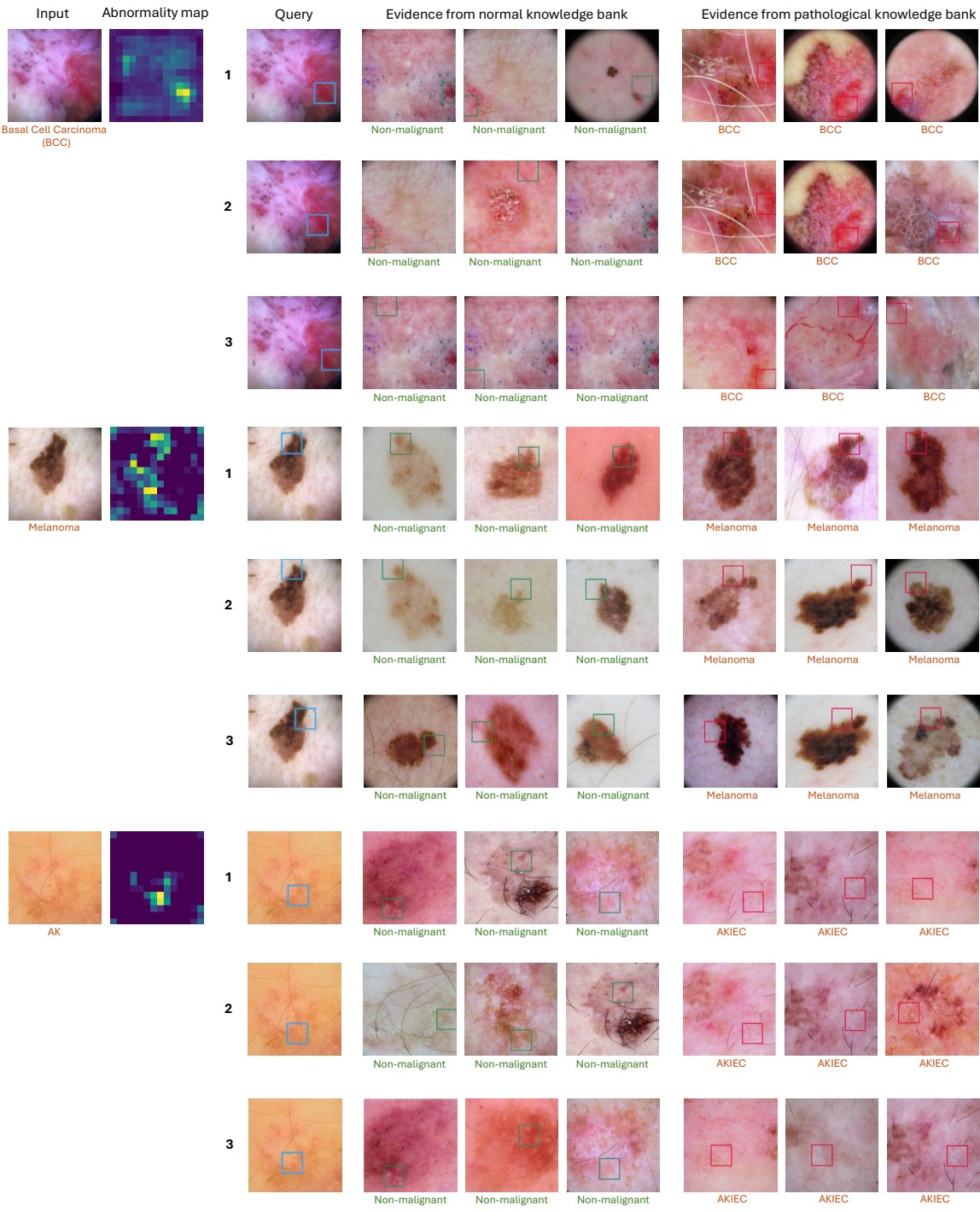

*Figure 9.* More examples of interpretability visualization for dermatology.

# D. More Experimental Results

In this section, we provide additional experimental results that are omitted from the main body due to space constraints.

## D.1. ROC Curves for Test Sets

Here, we present ROC curves for distinguishing normal and pathological samples on the entire test sets.

**(a)** **(b)**

**(c)** **(d)**

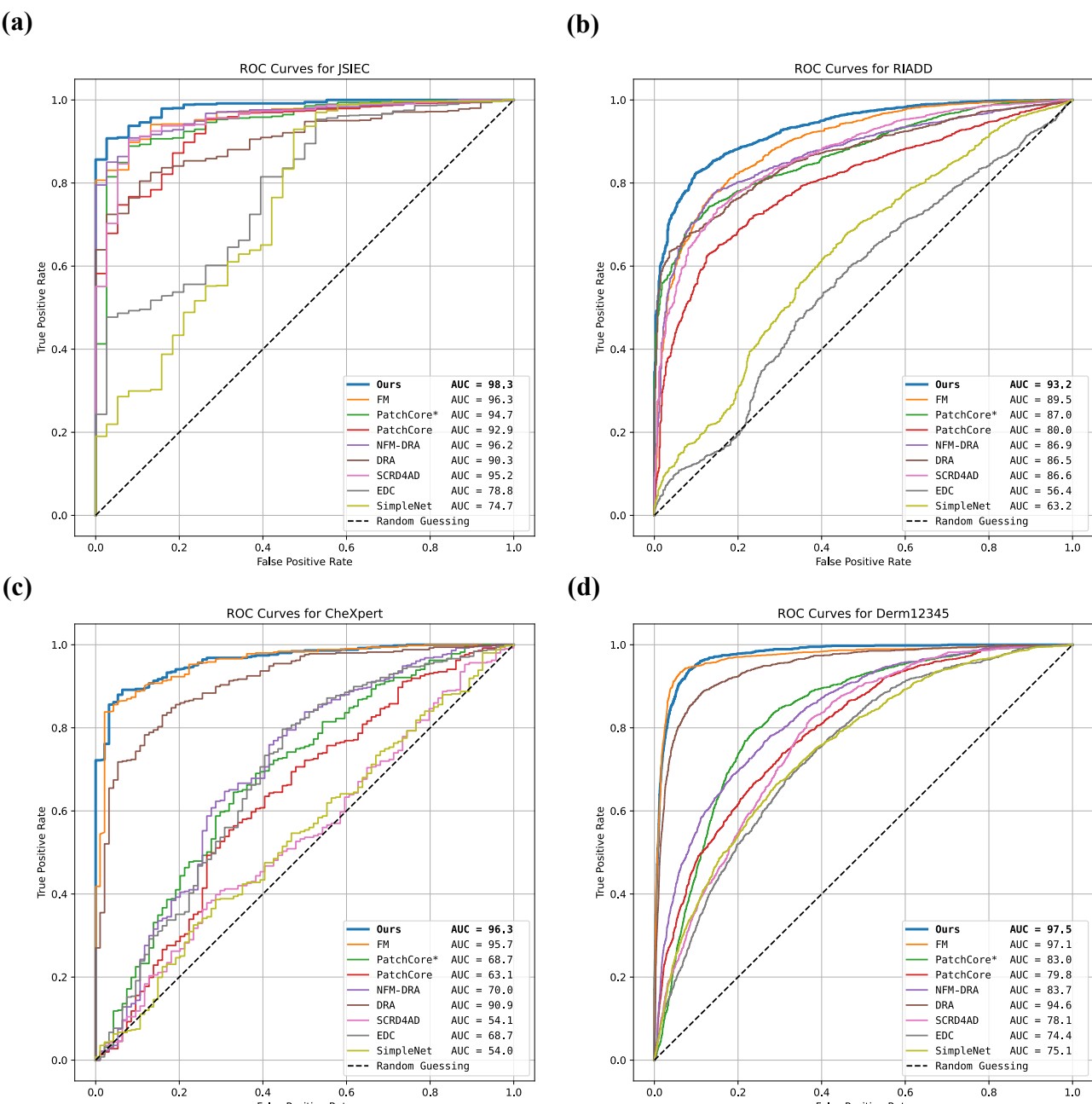

*Figure 10.* ROC curves for test sets: (a) JSIEC, (b) RIADD, (c) CheXpert, and (d) Derm12345.

## D.2. Category-Level Performance of Disease Screening

The category-level performance for disease screening is presented, based on evaluation with the test sets.

### D.2.1. CATEGORY-LEVEL PERFORMANCE ON JSIEC

*Table 3.* Category-level results for disease screening on JSIEC regarding **Spe@100%R (%)**. The **best** results are in bold, and the second-best results are underlined. * A variant of PatchCore that uses the same foundation models as ours.

| Category | Ours | FM | PatchCore* | PatchCore | NFM-DRA | DRA | SCRD4AD | EDC | SimpleNet | CIPL |
|---|---|---|---|---|---|---|---|---|---|---|
| Mean | **91.27** | 78.39 | 82.34 | 73.75 | 82.96 | 71.12 | 78.46 | 42.11 | 49.10 | 78.60 |
| Tessellated fundus | 44.74 | 15.79 | **71.05** | 36.84 | 36.84 | 63.16 | 60.53 | 13.16 | 28.95 | 0.00 |
| Large optic cup | 44.74 | 15.79 | 7.89 | 52.63 | 50.00 | 5.26 | 47.37 | 50.00 | 50.00 | **60.53** |
| DR1 | 50.00 | **57.89** | 10.53 | 2.63 | 2.63 | 13.16 | 13.16 | 5.26 | 2.63 | 2.63 |
| DR2 | **100.00** | **100.00** | 50.00 | 31.58 | 73.68 | **100.00** | 13.16 | 0.00 | 28.95 | **100.00** |
| DR3 | **100.00** | **100.00** | 92.11 | 50.00 | 94.74 | **100.00** | 63.16 | 5.26 | 47.37 | **100.00** |
| Possible glaucoma | **100.00** | **100.00** | 92.11 | 73.68 | 78.95 | 63.16 | 86.84 | 5.26 | 47.37 | 94.74 |
| Optic atrophy | **97.37** | 36.84 | 76.32 | 71.05 | 73.68 | 50.00 | 89.47 | 47.37 | 47.37 | 92.11 |
| Severe hypertensive retinopathy | **100.00** | **100.00** | 89.47 | 97.37 | **100.00** | **100.00** | 92.11 | 47.37 | 42.11 | **100.00** |
| Disc swelling and elevation | 78.95 | 76.32 | 92.11 | 76.32 | 73.68 | 5.26 | 76.32 | 55.26 | 52.63 | 39.47 |
| Dragged Disc | **100.00** | 92.11 | 97.37 | **100.00** | **100.00** | 78.95 | 97.37 | 47.37 | 50.00 | 86.84 |
| Congenital disc abnormality | **100.00** | 92.11 | 97.37 | **100.00** | **100.00** | 97.37 | 97.37 | 47.37 | 52.63 | 97.37 |
| Retinitis pigmentosa | **100.00** | 76.32 | 97.37 | 65.79 | 86.84 | **100.00** | 94.74 | 42.11 | 55.26 | 81.58 |
| Bietti crystalline dystrophy | **100.00** | **100.00** | 97.37 | 73.68 | 97.37 | **100.00** | **100.00** | 47.37 | 55.26 | 97.37 |
| Peripheral retinal degeneration and break | **100.00** | 10.53 | 97.37 | 94.74 | **100.00** | 50.00 | 97.37 | 60.53 | 55.26 | 73.68 |
| Myelinated nerve fiber | **100.00** | 86.84 | 97.37 | **100.00** | **100.00** | 50.00 | 84.21 | 52.63 | 52.63 | 86.84 |
| Vitreous particles | **100.00** | **100.00** | 97.37 | 94.74 | **100.00** | **100.00** | **100.00** | 63.16 | 55.26 | **100.00** |
| Fundus neoplasm | **100.00** | 92.11 | 97.37 | 76.32 | 92.11 | 55.26 | 94.74 | 47.37 | 55.26 | 81.58 |
| BRVO | **100.00** | **100.00** | **100.00** | 76.32 | 94.74 | 97.37 | 65.79 | 28.95 | 42.11 | **100.00** |
| CRVO | **100.00** | **100.00** | 92.11 | 73.68 | 92.11 | **100.00** | 94.74 | 60.53 | 52.63 | 97.37 |
| Massive hard exudates | **100.00** | **100.00** | 97.37 | **100.00** | **100.00** | **100.00** | **100.00** | 50.00 | 68.42 | **100.00** |
| Yellow-white spots-flecks | 92.11 | **100.00** | 44.74 | 73.68 | **100.00** | **100.00** | 73.68 | 47.37 | 52.63 | 97.37 |
| Cotton-wool spots | **100.00** | **100.00** | 71.05 | 78.95 | 92.11 | 89.47 | 65.79 | 47.37 | 55.26 | 97.37 |
| Vessel tortuosity | 44.74 | 7.89 | 68.42 | **76.32** | 73.68 | 50.00 | 47.37 | 47.37 | 42.11 | 68.42 |
| Chorioretinal atrophy-coloboma | **100.00** | **100.00** | 97.37 | 94.74 | **100.00** | **100.00** | 97.37 | 47.37 | 50.00 | 97.37 |
| Preretinal hemorrhage | **100.00** | **100.00** | 97.37 | **100.00** | **100.00** | 89.47 | 94.74 | 47.37 | 57.89 | 57.89 |
| Fibrosis | **100.00** | 92.11 | 97.37 | 81.58 | 92.11 | 92.11 | 97.37 | 60.53 | 57.89 | 94.74 |
| Laser Spots | **100.00** | **100.00** | 97.37 | 97.37 | **100.00** | **100.00** | **100.00** | 50.00 | 63.16 | 97.37 |
| Silicon oil in eye | **100.00** | 36.84 | 94.74 | **100.00** | **100.00** | 92.11 | **100.00** | 47.37 | 52.63 | 97.37 |
| Blur fundus without PDR | 84.21 | 36.84 | **92.11** | 81.58 | 81.58 | 0.00 | 86.84 | 0.00 | 47.37 | 28.95 |
| Blur fundus with suspected PDR | **100.00** | **100.00** | 97.37 | 63.16 | 97.37 | 63.16 | 47.37 | 23.68 | 44.74 | 39.47 |
| RAO | 86.84 | **100.00** | 97.37 | 73.68 | 71.05 | 0.00 | 47.37 | 52.63 | 57.89 | 76.32 |
| Rhegmatogenous RD | 92.11 | 57.89 | **97.37** | 73.68 | 76.32 | 34.21 | 92.11 | 50.00 | 50.00 | 50.00 |
| CSCR | **86.84** | 76.32 | 71.05 | 55.26 | 71.05 | 21.05 | 73.68 | 60.53 | 52.63 | 73.68 |
| VKH disease | 86.84 | 60.53 | 86.84 | 65.79 | 73.68 | **89.47** | 84.21 | 52.63 | 52.63 | 76.32 |
| Maculopathy | **100.00** | **100.00** | 52.63 | 65.79 | 78.95 | 86.84 | 92.11 | 47.37 | 52.63 | 86.84 |
| ERM | **86.84** | **86.84** | 71.05 | 34.21 | 50.00 | 81.58 | 78.95 | 47.37 | 44.74 | 81.58 |
| MH | **92.11** | 71.05 | 47.37 | 47.37 | 47.37 | 84.21 | 39.47 | 47.37 | 42.11 | 76.32 |
| Pathological myopia | **100.00** | **100.00** | 97.37 | 92.11 | **100.00** | **100.00** | 94.74 | 47.37 | 47.37 | 97.37 |

### D.2.2. CATEGORY-LEVEL PERFORMANCE ON RIADD

*Table 4.* Category-level results for disease screening on RIADD regarding **Spe@100%R (%)**. The **best** results are in bold, and the second-best results are underlined. * A variant of PatchCore that uses the same foundation models as ours.

| Category | Ours | FM | PatchCore* | PatchCore | NFM-DRA | DRA | SCRD4AD | EDC | SimpleNet | CIPL |
|---|---|---|---|---|---|---|---|---|---|---|
| Mean | **55.35** | 51.75 | 41.63 | 29.36 | 37.64 | 29.74 | 43.98 | 9.08 | 14.26 | 43.99 |
| DR | **69.81** | 58.30 | 10.76 | 2.24 | 1.94 | 10.91 | 20.63 | 0.75 | 0.00 | 24.22 |
| ARMD | **77.13** | 60.99 | 23.32 | 28.10 | 47.09 | 45.89 | 47.23 | 1.64 | 2.39 | 62.93 |
| MH | 11.36 | 3.14 | **31.69** | 11.21 | 15.10 | 0.00 | 8.97 | 0.15 | 2.99 | 2.09 |
| DN | 37.22 | **37.67** | 0.60 | 0.00 | 1.35 | 2.24 | 1.49 | 0.30 | 0.00 | 19.28 |
| MYA | 69.81 | 44.54 | 18.39 | 42.30 | 57.55 | **73.54** | 60.69 | 9.12 | 9.72 | 28.25 |
| BRVO | **91.18** | 76.38 | 48.88 | 27.20 | 23.77 | 20.33 | 52.02 | 3.74 | 15.25 | 39.46 |
| TSLN | 3.29 | 13.45 | 5.08 | **16.14** | 13.30 | 3.74 | 5.23 | 6.73 | 0.75 | 10.16 |
| ERM | 27.06 | 30.34 | 3.59 | 6.58 | 8.22 | 22.12 | **71.45** | 1.05 | 4.78 | 66.82 |
| LS | **97.91** | 94.32 | 43.65 | 14.20 | 19.13 | 28.70 | 52.77 | 1.94 | 32.44 | 80.87 |
| MS | 90.28 | **91.33** | 41.70 | 50.82 | 74.14 | 86.55 | 58.89 | 28.25 | 6.43 | 31.69 |
| CSR | 20.78 | **47.09** | 22.72 | 1.05 | 2.09 | 0.00 | 32.29 | 3.29 | 1.20 | 16.29 |
| ODC | 7.47 | 4.48 | 0.15 | 0.15 | 0.15 | 0.75 | 1.64 | 0.75 | 0.75 | **16.14** |
| CRVO | 84.45 | 87.44 | **88.94** | 23.92 | 41.26 | 65.92 | 66.52 | 4.19 | 22.87 | 81.61 |
| TV | 82.81 | 90.28 | **99.40** | 77.28 | 74.14 | 12.71 | 86.55 | 75.64 | 23.32 | 62.48 |
| AH | 98.80 | 88.79 | 98.65 | 40.36 | 94.32 | 98.65 | 97.31 | 0.30 | 31.39 | **100.00** |
| ODP | 25.11 | 7.77 | 5.38 | 0.75 | 0.60 | 3.44 | **26.16** | 0.30 | 11.06 | 6.13 |
| ODE | 22.12 | **24.96** | 7.47 | 13.15 | 18.54 | 1.94 | 18.09 | 10.16 | 9.72 | 14.35 |
| ST | 59.79 | 8.97 | **73.99** | 54.41 | 52.47 | 35.13 | 25.56 | 26.76 | 23.32 | 33.48 |
| AION | 63.08 | 47.68 | **98.21** | 69.21 | 82.21 | 2.99 | 50.67 | 4.04 | 16.29 | 29.00 |
| PT | 48.88 | **62.48** | 19.58 | 5.98 | 4.63 | 0.75 | 22.42 | 11.06 | 4.78 | 36.62 |
| RT | 99.40 | 90.28 | **100.00** | 94.17 | 95.37 | 77.73 | 97.31 | 4.19 | 51.27 | 76.83 |
| RS | **69.96** | 64.87 | 61.88 | 60.84 | 60.39 | 9.42 | 26.01 | 2.24 | 9.72 | 39.91 |
| CRS | 35.28 | 41.85 | 26.91 | 50.07 | 67.12 | 1.49 | **70.55** | 18.98 | 27.20 | 48.13 |
| EDN | 98.95 | 98.21 | 48.13 | 30.04 | 82.51 | 99.25 | 81.61 | 10.16 | 46.19 | **99.85** |
| RPEC | 24.66 | **50.22** | 17.79 | 0.00 | 0.00 | 28.40 | 20.78 | 2.24 | 6.58 | 35.28 |
| MHL | 29.75 | 45.59 | 12.41 | 6.73 | 4.93 | 14.80 | 28.55 | 22.42 | 3.74 | **60.99** |
| RP | 96.86 | 41.41 | 95.96 | 77.43 | 90.43 | 82.96 | 94.77 | 1.35 | 30.34 | **98.80** |
| OTHER | 6.58 | 36.17 | **60.54** | 17.64 | 21.08 | 2.24 | 5.23 | 2.39 | 4.93 | 10.01 |

### D.2.3. CATEGORY-LEVEL PERFORMANCE ON CHEXPERT

*Table 5.* Category-level results for disease screening on CheXpert regarding **Spe@100%R (%)**. The **best** results are in bold, and the second-best results are underlined. * A variant of PatchCore that uses the same foundation models as ours.

| Category | Ours | FM | PatchCore* | PatchCore | NFM-DRA | DRA | SCRD4AD | EDC | SimpleNet | CIPL |
|---|---|---|---|---|---|---|---|---|---|---|
| Mean | **68.72** | 60.00 | 15.53 | 11.06 | 13.62 | 18.72 | 5.53 | 14.89 | 8.94 | 49.15 |
| Atelectasis | 28.72 | **35.11** | 8.51 | 13.83 | 17.02 | 28.72 | 3.19 | 0.00 | 7.45 | 30.85 |
| Cardiomegaly | **34.04** | 25.53 | 11.70 | 2.13 | 5.32 | 9.57 | 4.26 | 4.26 | 2.13 | 12.77 |
| Consolidation | **100.00** | **100.00** | 34.04 | 10.64 | 17.02 | 20.21 | 17.02 | 26.60 | 23.40 | 55.32 |
| Edema | **85.11** | 77.66 | 11.70 | 2.13 | 5.32 | 26.60 | 0.00 | 15.96 | 4.26 | 82.98 |
| Pleural Effusion | **95.74** | 61.70 | 11.70 | 26.60 | 23.40 | 8.51 | 3.19 | 27.66 | 7.45 | 63.83 |

### D.2.4. CATEGORY-LEVEL PERFORMANCE ON DERM12345

*Table 6.* Category-level results for disease screening on Derm12345 regarding **Spe@100%R (%)**. The **best** results are in bold, and the second-best results are underlined. * A variant of PatchCore that uses the same foundation models as ours.

| Category | Ours | FM | PatchCore* | PatchCore | NFM-DRA | DRA | SCRD4AD | EDC | SimpleNet | CIPL |
|---|---|---|---|---|---|---|---|---|---|---|
| Mean | **78.49** | 55.41 | 46.90 | 36.82 | 41.80 | 56.91 | 32.86 | 22.15 | 20.94 | 54.95 |
| ALM | 31.30 | 22.63 | **34.16** | 21.54 | 20.86 | 27.74 | 22.07 | 9.92 | 4.58 | 33.28 |
| ANM | **91.94** | 68.56 | 76.60 | 69.79 | 82.10 | 89.60 | 31.24 | 49.97 | 31.80 | 89.64 |
| LM | **59.07** | 0.90 | 5.40 | 23.26 | 29.05 | 33.60 | 30.50 | 2.01 | 0.92 | 42.55 |
| LMM | 96.81 | **96.99** | 43.44 | 60.64 | 74.15 | 95.31 | 57.90 | 11.86 | 53.83 | 93.49 |
| MEL | **29.71** | 22.43 | 0.33 | 12.68 | 11.92 | 15.53 | 0.61 | 8.20 | 8.59 | 20.20 |
| AK | **75.85** | 3.56 | 17.72 | 1.29 | 3.19 | 17.39 | 14.07 | 13.71 | 1.01 | 12.02 |
| BCC | **82.05** | 31.79 | 22.92 | 3.17 | 3.56 | 0.05 | 0.99 | 6.40 | 1.27 | 8.54 |
| BD | **95.49** | 92.86 | 22.19 | 13.79 | 13.27 | 49.20 | 38.73 | 2.34 | 2.66 | 68.96 |
| CH | **84.04** | 16.55 | 81.34 | 65.81 | 73.93 | 79.24 | 43.27 | 16.96 | 10.38 | 49.02 |
| MPD | **99.29** | 99.25 | 82.60 | 51.31 | 64.02 | 98.78 | 40.27 | 67.65 | 67.11 | 93.77 |
| SCC | **81.20** | 70.65 | 64.24 | 22.93 | 22.38 | 57.57 | 46.55 | 17.70 | 23.62 | 12.60 |
| DFSP | **98.78** | 97.72 | 92.56 | 90.42 | 93.48 | 97.60 | 74.72 | 62.81 | 53.45 | 97.28 |
| KS | 94.83 | **96.48** | 66.12 | 42.00 | 51.47 | 78.25 | 26.25 | 18.42 | 12.94 | 92.99 |

## D.3. Deployment Efficiency

We report the deployment cost of the dual knowledge banks. The ophthalmology, radiology, and dermatology banks contain 256k, 1,024k, and 196k vectors, respectively, with corresponding memory footprints of 1,002 MB, 4,004 MB, and 766 MB. In the largest radiology setting, the dual knowledge banks contain 1,024k vectors, require 4,004 MB of memory, and take approximately 0.6 s for retrieval, indicating that the evidence retrieval process remains lightweight for deployment.

## D.4. Stress Test on Pathological Bank Contamination

The pathological knowledge bank inevitably contains normal regions because image-level pathological labels do not provide patch-level annotations. To evaluate the robustness of our framework to such contamination, we conduct a stress test on JSIEC by injecting 20% normal samples into the pathological bank during construction. We also evaluate a SoftPatch-like (Jiang et al., 2022) denoising strategy during knowledge bank construction. As shown in the results below, our model maintains stable performance under stress testing on the JSIEC dataset, and explicit denoising yields comparable scores.

*Table 7.* Stress test and denoising results on JSIEC (%).

| Metric | Original | Stress test | Denoising strategy |
|---|---|---|---|
| AUROC | 98.06 | 97.71 | 96.90 |
| Spe@99%R | 91.62 | 90.10 | 92.11 |

## D.5. Quantitative Interpretability Evaluation

We evaluate localization interpretability using expert-annotated lesion masks from a reader study on 20 fundus images. As shown in Table 8, our approach achieves more accurate localization than PatchCore* and CIPL.

*Table 8.* Reader study results for localization interpretability.

| Method | Dice | IoU |
|---|---|---|
| PatchCore* | $0.45 \pm 0.13$ | $0.30 \pm 0.12$ |
| CIPL | $0.58 \pm 0.16$ | $0.42 \pm 0.16$ |
| Ours | $\mathbf{0.66 \pm 0.06}$ | $\mathbf{0.50 \pm 0.07}$ |

