# OpenReview forum: "Evidential Reasoning Advances Interpretable Real-World Disease Screening"
_ICML.cc/2026/Conference — ICML 2026 regular_

### Official Review · Reviewer_SxcF · 2026-03-09

**Soundness:** 2
**Presentation:** 3
**Significance:** 2
**Originality:** 3
**Overall Recommendation:** 3
**Confidence:** 5

**Summary:**

The paper proposes EviScreen, an interpretable framework for disease screening. By providing retrospection interpretability through regional evidence retrieved from dual knowledge banks, EviScreen enables evidence-aware reasoning, allowing not only accurate diagnostic performance but also advanced localization interpretability through abnormality maps generated via contrastive retrieval.

**Compliance With Llm Reviewing Policy:**

Affirmed.

**Final Justification:**

The authors completely overlooks existing advances in interpretable medical image analysis in their paper, where prior work has already addressed the problem of providing precise localization based interpretability while performing disease analysis, not only for screening tasks (binary classification) but also for multi class and even multi label disease prediction.

While the authors clarify the distinctions between CIPL and EviScreen in their rebuttal, their comparison remains limited to performance and computational efficiency, with no meaningful assessment of interpretability. To some extent, I believe that these performance improvements may largely rely on the strong capabilities of the foundation model.

For these reasons, I would suggest rejection in its current form. A more thorough investigation of recent advances in interpretable medical image analysis, together with a more comprehensive discussion of these works in the paper, would help strengthen the manuscript.

**Key Questions For Authors:**

See Weaknesses.

**Limitations:**

yes

**Strengths And Weaknesses:**

**Strengths**
1. The paper is well written and well structured.
3. The paper provides comprehensive experiments across ten public datasets covering three critical medical modalities.
2. The paper evaluates the model using cross-evaluation (by overlapping the training&validation and testing sets), demonstrating its effectiveness in real-world disease screening.

**Weaknesses**
1. The work appears to focus more on anomaly detection rather than interpretable disease diagnosis. The paper does not compare the proposed method with other related explainable models, making it unclear whether the interpretability of the method is effective.

2. It is unclear in the paper why visual evidence from historical cases is crucial for explaining the model's predictions. As far as I know, current popular self-explaining medical models, such as Xprotonet [1], are already capable of providing explanations for different diseases in medical images by considering both normal and pathological regions based on training samples [2,3]. However, the paper overlooks existing contributions.

3. The paper claims to be interpretable, and although it provides visualizations for explanation, it fails to demonstrate the model's performance on general datasets. There is a lack of evaluation using interpretability metrics, such as fidelity or faithfulness scores.

[1] Xprotonet: diagnosis in chest radiography with global and local explanations (cvpr'21)

[2] Cross-and intra-image prototypical learning for multi-label disease diagnosis and interpretation. (TMI'25)

[3] Parsecaps: An interpretable parsing capsule network for medical image diagnosis. (AAAI'25)

---

> ### Author Rebuttal · Authors · 2026-03-31
>
> We sincerely appreciate your constructive feedback and positive recognition of our "well written" presentation, "comprehensive experiments", and "effectiveness in real-world disease screening". Your insightful comments have been integrated into our revision, and we address your specific concerns below.
> ***
> ### Response to comment-1:
> > Concerns about the research scope and comparison with other explainable models.
>
> Thank you for your valuable comments.
> - We respectfully clarify that the ***primary scope*** of this work is ***disease screening***. While screening is formulated similarly to anomaly detection, it holds critical clinical significance. It serves as a vital and distinct step prior to ***disease diagnosis***, designed to identify abnormalities within an apparently healthy population [1].
>
> - ***Comparison with other explainable methods.*** Following your literature recommendation, we added experiments comparing our method against a SOTA prototype-based model (CIPL, TMI'25 [2]). The table below shows that our method consistently outperforms CIPL, often by substantial margins (e.g., achieving 91.27% Spe@100%R on JSIEC vs. CIPL's 78.60%, an improvement of over 12%).
>
>   |Dataset|Method|Spe@100%R|Spe@99%R|Spe@95%R|CSR|AUROC|Average Precision|
>   |:-:|:-:|:-:|:-:|:-:|:-:|:-:|:-:|
>   |JSIEC|**Ours**|**91.27**|**91.62**|**94.71**|**88.95**|**98.06**|**96.10**|
>   ||CIPL|78.60|79.57|87.33|73.41|94.83|91.36|
>   |RIADD|**Ours**|**55.35**|**59.39**|**72.92**|**54.38**|**91.32**|**60.35**|
>   ||CIPL|43.99|51.40|64.92|44.85|87.09|59.08|
>   |CheXpert|**Ours**|**68.72**|**74.26**|**84.04**|**70.59**|**96.72**|**94.71**|
>   ||CIPL|43.83|48.72|68.09|47.29|92.57|84.60|
>   |Derm12345|**Ours**|**78.49**|**85.78**|**90.29**|**77.92**|**97.43**|**34.23**|
>   ||CIPL|54.95|68.08|82.04|54.74|94.00|13.37|
>
> [1] Principles and practice of screening for disease. World Health Organization. 1968.
>
> [2] Cross-and intra-image prototypical learning for multi-label disease diagnosis and interpretation. TMI'25.
> ***
> ### Response to comment-2:
> > Concerns about the importance of visual evidence and contributions of prototype-based models.
>
> Thank you for your insightful comments. We have incorporated the following important discussions into the revised manuscript:
>
> - **Why visual evidence from historical cases is crucial:**
>
>     In clinical practice, decision-making is inherently grounded in evidence-based medicine. Providing explicit visual evidence makes AI decision-making more transparent and trustworthy. Notably, prior works such as CIPL [2] (one of the valuable papers you recommended) similarly emphasize that historical case-based visual evidence ***"provides a transparent process of disease interpretation"***.
>
> - **Distinct advantages of our dual knowledge banks over prototypes:**
>
>   1. ***Computational scalability:*** Prototype-based models require ***computationally intensive training*** to learn prototypes. In contrast, our dual knowledge banks are constructed ***training-free*** using features from a frozen foundation model.
>   2.  ***Capacity scalability:*** Prototype models suffer from ***capacity bottlenecks***. For instance, Figure 6 of CIPL [2] shows a performance ***decline*** when exceeding 50 prototypes per class. Conversely, Figure 5(b) of our manuscript shows our method can scale to ***over 200k vectors*** (10,000 samples).
>   3. ***Notable performance improvements:*** Benefiting from this scalability, our method preserves a broader range of visual patterns, resulting in significant performance gains (detailed in *Response to comment-1*).
> ***
> ### Response to comment-3:
> > Concerns about general datasets and interpretability metrics.
>
> Thank you for this constructive feedback. We respectfully clarify that our evaluation framework is comprehensive, covering ***ten diverse public datasets*** across ***three major medical modalities***. Crucially, we employed cross-dataset evaluation to verify the robust generalization on general real-world data.
>
> ***Quantitative interpretability metrics:***
> - Following your helpful suggestion, we ***quantified faithfulness*** using the widely-recognized Average Drop metric introduced in Grad-CAM++ [3] (over 4,600 citations). This metric measures the decrease in prediction confidence when the input is masked by the explanation map. On JSIEC, our method achieves an Average Drop of 26.5% (vs. 36.0% for Grad-CAM++; lower is better), showing the faithfulness of our interpretation.
> - In addition, since existing prototype-based models often evaluate localization instead of faithfulness, we conducted a ***reader study*** measuring spatial localization accuracy (detailed in *Response to comment-6* of `Reviewer we2G`).
>
> [3] Grad-cam++: Generalized gradient-based visual explanations for deep convolutional networks. WACV'18.
>
> ***
> Thank you again for your time and expertise, which have greatly improved the quality of our manuscript. If you have further questions, please do not hesitate to let us know.
> ***

---

> > ### Author Rebuttal · Reviewer_SxcF · 2026-04-03
> >
> > Thank the authors for their rebuttal. However, based on their response, I confirm that this paper is primarily focused on anomaly detection rather than interpretable medical image analysis. Although the authors argue that the method is designed for disease screening, this appears to be more of an application scenario than a technical contribution.
> >
> > More importantly, the authors **completely overlooks existing advances in interpretable medical image analysis in their paper**, where prior work has already addressed the problem of providing precise localization based interpretability while performing disease analysis, not only for screening tasks (binary classification) but also for multi class and even multi label disease prediction.
> >
> > In addition, as stated in the authors’ response, the idea emphasized in EviScreen, i.e. "visual evidence from historical cases is crucial for explaining model predictions," has already been explored in previous research. Therefore, I find the significance of this paper to be rather limited.
> >
> > The paper does not discuss prior progress in interpretable medical image analysis, nor does it provide meaningful interpretability comparisons. While the authors clarify the distinctions between CIPL and EviScreen in their rebuttal, their comparison remains limited to performance and computational efficiency, with no meaningful assessment of interpretability, such as evaluating whether the predicted evidence is accurately localized using quantitative metrics like IoU. To some extent, I believe that these classification performance improvements largely rely on the strong capabilities of the foundation model.
> >
> > For these reasons, I would suggest rejection in its current form. A more thorough investigation of recent advances in interpretable medical image analysis, together with a more comprehensive discussion of these works in the paper, would help strengthen the manuscript.

---

> > > ### Author Response · Authors · 2026-04-04
> > >
> > > Dear Reviewer SxcF,
> > >
> > > Thank you sincerely for your continued engagement with our work. Your comments have been genuinely constructive in improving the quality of our manuscript, and we would like to respectfully address your remaining concerns below.
> > >
> > > ---
> > >
> > > ***1. Quantitative evaluation of interpretability***
> > >
> > > Following your constructive suggestion, we provide a quantitative localization evaluation using expert-annotated masks from the reader study conducted during the rebuttal period. Results show that EviScreen produces more accurate localization than CIPL, a state-of-the-art prototype-based interpretable model:
> > >
> > > |Method|Dice|IoU|
> > > |:-:|:-:|:-:|
> > > |**Ours**|**0.66±0.06**|**0.50±0.07**|
> > > |CIPL|0.58±0.16|0.42±0.16|
> > >
> > > ***2. Clarification on our primary scope: disease screening***
> > >
> > > We respectfully clarify that disease screening is a clinically distinct task that goes ***beyond anomaly detection***. Anomaly detection methods rely on normal data during training and cannot leverage pathological historical cases. In contrast, our method provides an effective paradigm for incorporating both normal and pathological cases, ***consistently outperforming*** anomaly detection methods, supervised fine-tuning, and prototype-based interpretable methods in our comparative experiments.
> > >
> > > ***3. Limitations of prior interpretable methods beyond computational efficiency, and our solution***
> > >
> > > We deeply respect the contributions of prototype-based methods such as CIPL and XProtoNet. However, these interpretable methods rely on manually predefined prototype classes and are ***capacity-bounded by design***.
> > >
> > > Specifically, the number of prototypes is fixed prior to training; notably, CIPL's own ablation (Figure 6) shows ***performance degrading*** beyond 50 prototypes per class. This capacity constraint is a critical ***clinical bottleneck rather than a mere engineering concern***. Real-world disease screening involves highly diverse pathological presentations. A model whose representational capacity saturates at a small fixed number of prototypes ***is unable*** to capture this diversity.
> > >
> > > EviScreen overcomes this limitation by replacing learned prototypes with ***highly scalable*** knowledge banks. As demonstrated in Figure 5(b), performance improves steadily up to 200k+ vectors with no sign of saturation. This mirrors a fundamental tenet of clinical practice: broader accumulated experience directly translates to superior judgment.
> > >
> > > We will include a comprehensive discussion and comparison with prior interpretable methods in the revised manuscript.
> > >
> > > ***4. Source of performance improvements***
> > >
> > > We wish to clarify that the performance gains of EviScreen are driven primarily by the proposed framework, rather than the underlying foundation model. As shown in Table 1, EviScreen consistently outperforms ***standard foundation model fine-tuning (FM)***, which utilizes the ***exact same foundation model*** with direct supervision, as well as ***PatchCore\****, an anomaly detection method built upon the identical foundation model. For instance, on the JSIEC dataset, EviScreen surpasses FM and PatchCore\* by 12.88\% and 8.93\% in Spe@100%R, respectively. These controlled comparisons confirm that the observed improvements stem directly from the EviScreen architecture itself, not merely from the capacity of the foundation models.
> > >
> > > ---
> > >
> > > We sincerely hope the above clarifications address your remaining concerns. We are fully committed to incorporating these insights into the revised manuscript, and we deeply appreciate the time and expertise you have dedicated to reviewing our paper.
> > >
> > > Best Regards,
> > >
> > > The Authors

---

### Official Review · Reviewer_we2G · 2026-03-11

**Soundness:** 3
**Presentation:** 3
**Significance:** 3
**Originality:** 3
**Overall Recommendation:** 4
**Confidence:** 3

**Summary:**

This paper proposes an evidence reasoning framework named EviScreen, aiming to address the issues of lack of interpretability and insufficient performance of existing medical image screening models in clinical practice. The core of this framework lies in the construction of Dual Knowledge Banks that contain the regional characteristics of normal and pathological cases. During the reasoning process, the model retrievals historical cases as "evidence" and makes decisions using the evidence-based enhanced attention mechanism, thereby achieving "retrospective Interpretability" similar to that of clinicians (Retrospection Interpretability). In addition, the author proposed a set of evaluation metrics that better meet clinical needs (such as Spe@X%R and CSR), and verified their superiority on 10 public datasets covering ophthalmology, radiology and dermatology.

**Compliance With Llm Reviewing Policy:**

Affirmed.

**Key Questions For Authors:**

1. What changes occur in the performance and retrieval efficiency of the model under basic models of different scales and different parameter quantities (such as ViT-S vs ViT-L)?

2. In the real clinical deployment scenarios, what is the update and maintenance mechanism of the dual knowledge base? If new disease categories are added, is it necessary to retrain the reasoning module?

3. How does the degree of overlap (Dice/IoU) of the "abnormal Map" (Abnormality Map) generated by comparative retrieval perform specifically when compared with the lesion labels of senior doctors?

**Limitations:**

Yes

**Strengths And Weaknesses:**

Strength:

1. Clinical decision simulation: Unlike traditional methods that rely solely on significance maps, EviScreen provides inferential evidence by citing historical cases, which not only improves accuracy but also enhances clinical trust.

2. Full mining of pathological information: Overcoming the limitation of previous Deviation-based models that could not effectively utilize pathological sample information, rich pathological patterns are learned through a dual knowledge base.

3. The evaluation system is rigorous and the experimental scope is wide: the study not only used internal validation but also introduced multiple external test sets and designed indicators such as Spe@95%R that directly reflect clinical costs (reducing misdiagnosis and unnecessary reexamination). It demonstrated significant performance improvements in all three types of medical imaging modalities, especially in CSR indicators, far exceeding the comparison methods.

Weakness:

1. Although Faiss acceleration is used, the retrieval delay and memory usage in an extremely large-scale knowledge base remain potential challenges that may be faced during real-time deployment

2. The performance of the framework is highly dependent on the quality of features extracted by the selected visual base model (such as RETFound, PanDerm).

3. The purity of pathological plaques: As the author stated, pathological images contain normal areas. Although contrast retrieval has been proposed for relief, the robustness under extremely subtle lesions still needs further exploration.

---

> ### Author Rebuttal · Authors · 2026-03-31
>
> We sincerely appreciate your constructive feedback and positive recognition of "enhancing clinical trust", "full mining of pathological information", and "evaluation system is rigorous". We have carefully incorporated your insightful comments into our revision. Below, we address your concerns point-by-point.
> ***
> ### Response to comment-1:
> > Concerns about real-time deployment.
>
> Thank you for your valuable comments. Empirical testing shows that for dual knowledge banks containing 1,024k vectors, the retrieval latency is approximately 0.6 seconds, with a memory footprint of just 4 GB. These low computational overheads are highly accessible for real-time clinical deployment.
> ***
> ### Response to comment-2:
> > Concerns about foundation model-based design.
>
> Thank you for your insightful comments. While our model leverages features extracted by foundation models, the performance improvement is driven primarily by the proposed evidential reasoning architecture. As shown in Table 1, our method consistently ***outperforms standard foundation model fine-tuning***.
>
> Moreover, in the current era of prevalent foundation models, our framework provides an ***effective paradigm for adapting foundation models*** to specialized disease screening. We hope this research inspires further exploration in this direction.
> ***
> ### Response to comment-3:
> > Concerns regarding the purity of pathological patches.
>
> While the pathological bank inevitably contains some normal features due to the lack of explicit patch-level labels, our method inherently mitigates this issue through two core mechanisms:
> 1. Normal patches are naturally distant from pathological queries in the latent space. Consequently, our ***evidence retrieval*** process naturally prioritizes selecting similar pathological features over random or normal patches.
> 2. During the ***evidence-aware reasoning*** phase, the cross-attention module dynamically learns to weigh suitable evidence, making the model inherently robust to potential noise.
>
> To validate this robustness, we designed a ***stress test*** where we manually injected 20% normal samples during the construction of the pathological bank. As shown in the table below, our model exhibits stable performance under the stress test in the JSIEC dataset, validating the robustness.
>
> ||Original|Stress testing|
> |:-:|:-:|:-:|
> |AUROC (%)|98.06|97.71|
> |Spe@99%R (%)|91.62|90.10|
> ***
> ### Response to comment-4:
> > Questions about performance and efficiency under different model scales.
>
> Thank you for your insightful question. To address this, we evaluated our framework using RAD-DINO [1] as a smaller-scale foundation model (ViT-B) to compare against the default scale (ViT-L). Based on the CheXpert test set results below, while the larger model yields slight performance improvements, our framework maintains highly competitive accuracy and retrieval efficiency at both backbone scales.
>
> |Backbone scale|Parameters|AUROC (%)|Retrieval latency (s)|
> |:-:|:-:|:-:|:-:|
> |ViT-B|86M|95.75|0.4|
> |ViT-L|307M|96.72|0.6|
>
> [1] Exploring scalable medical image encoders beyond text supervision. Nature Machine Intelligence. 2025.
> ***
> ### Response to comment-5:
> > Questions about the update and maintenance mechanism.
>
> Thank you for your insightful question. The dual knowledge bank design is ***flexible***, and new data can be appended without the need to reconstruct the bank from scratch. Importantly, our model exhibits inherent generalizability to ***unseen*** disease categories. As shown in the table below, our model achieves high Spe@100%R (%) scores on representative unseen categories within the JSIEC dataset.
>
> |Category|CSCR|MH|Retinitis pigmentosa|Optic atrophy|Fundus neoplasm|
> |:-:|:-:|:-:|:-:|:-:|:-:|
> |Ours|86.84|92.11|100.00|97.37|100.00|
> |Top-performing Baseline|78.32|84.21|100.00|89.47|97.37|
>
> Although training on a broader range of data may further enhance performance, it is not necessary to retrain the reasoning module to handle new diseases. We will expand the discussion on maintenance protocols in the revised manuscript to prevent any reader misunderstanding.
> ***
> ### Response to comment-6:
> > Questions about the localizability of abnormality maps.
>
> Thank you for your valuable question. To quantitatively assess the spatial accuracy of our abnormality maps, we conducted an expert-based ***reader study***. We invited a senior doctor to annotate the lesion regions of 20 fundus images from the test sets. As shown in the table below, our method produces more precise abnormality maps than the comparative method (PatchCore), further validating its reliable localizability.
>
> |Method|Dice|IoU|
> |:-:|:-:|:-:|
> |**Ours**|**0.66±0.06**|**0.50±0.07**|
> |PatchCore|0.45±0.13|0.30±0.12|
> ***
> Again, we sincerely appreciate your time and effort in reviewing. Revisions following your comments have greatly strengthened our manuscript. If you have further questions, please do not hesitate to let us know.
> ***

---

> > ### Author Rebuttal · Reviewer_we2G · 2026-04-04
> >
> > Thank you to the authors for their detailed rebuttal and clarifications. I have read the response and appreciate the additional explanations provided. After considering the rebuttal, I believe my main concerns have been adequately addressed to the extent possible, but my overall assessment of the paper remains unchanged. Therefore, I decide to maintain my original score.

---

> > > ### Author Response · Authors · 2026-04-06
> > >
> > > Dear Reviewer we2G,
> > >
> > > Thank you for taking the time to carefully read our rebuttal. We are ***truly encouraged by your positive assessment*** of our work!
> > >
> > > We are genuinely glad to hear that your ***main concerns have been adequately addressed to the extent possible***. Your critical perspective has been invaluable in helping strengthen our manuscript, and we will incorporate your feedback into the revised version of the manuscript.
> > >
> > > Once again, thank you for the time and effort you devoted to reviewing our work.
> > >
> > > Best regards,
> > >
> > > The Authors

---

### Official Review · Reviewer_Zae3 · 2026-03-12

**Soundness:** 3
**Presentation:** 3
**Significance:** 2
**Originality:** 2
**Overall Recommendation:** 4
**Confidence:** 3

**Summary:**

This paper proposes EviScreen, a retrieval- and evidence-aware framework for interpretable disease screening from medical images. The method builds dual knowledge banks of regional features from normal and pathological historical cases, retrieves per-patch evidence at inference, and performs evidence-aware cross-attention and inter-patch self-attention to predict disease while exposing explicit, case-referenced “retrospection interpretability.” A training-free variant generates abnormality maps by contrasting distances to the two banks, yielding improved localization interpretability. Across 10 datasets spanning ophthalmology, radiology, and dermatology, the authors report strong gains, particularly on clinically oriented metrics they introduce: specificity at fixed high recall and a clear separation rate.

**Compliance With Llm Reviewing Policy:**

Affirmed.

**Key Questions For Authors:**

1. How large are the dual knowledge banks after coreset subsampling for each modality (number of vectors, memory footprint)？
2. How robust is the pathological bank to patch-level contamination with normal tissue? Have you tried soft weighting or denoising strategies (e.g., SoftPatch-like) or cosine distance normalization to reduce noise?

**Limitations:**

yes

**Strengths And Weaknesses:**

## Strengths

1. Proposes a training‑free contrastive retrieval that compares distances to the two banks, yielding abnormality maps and decisions without additional learning.
2. Evaluates across multiple modalities and datasets in three clinical domains, with external test sets to assess generalization.
3. Visualizations effectively convey retrospection and localization.


## Weaknesses

1. The pathological knowledge bank is constructed from whole-pathology images without patch-level labels; thus, many “pathological” bank entries are likely normal tissue. While the contrastive design partially mitigates this, no explicit mechanism disentangles normal vs abnormal patches within positive cases in the main (trained) model.
2. Memory and compute costs of dual banks and k-NN retrieval at scale are not quantified (bank size per modality, latency, GPU/CPU RAM, query throughput), raising concerns for deployment.
3. Some baseline choices are limited. Prototype/case-based interpretability methods and recent memory-free AD models are not included in the comparisons.

---

> ### Author Rebuttal · Authors · 2026-03-31
>
> We sincerely appreciate your constructive feedback and positive comments regarding our methodology, experiments, and visualizations ("effectively convey retrospection and localization"). We have carefully incorporated your insightful feedback into our revision. Below, we address your concerns point-by-point.
> ***
> ### Response to comment-1:
> > Concerns about contamination of normal tissues in the pathological bank.
>
> Thank you for your valuable comments. While the pathological bank inevitably contains some normal features due to the lack of explicit patch-level labels, our method ***inherently mitigates*** this issue through two core mechanisms:
> 1. Normal patches are naturally distant from pathological queries in the latent space. Consequently, our ***evidence retrieval*** process naturally prioritizes selecting similar pathological features over random or normal patches.
>
> 2. During the ***evidence-aware reasoning*** phase, the cross-attention module dynamically learns to weight retrieved evidence appropriately, making the model inherently robust to potential noise.
>
> To explicitly validate this robustness, we have conducted a dedicated ***stress test*** (detailed in *Response to comment-5*).
>
> ***
> ### Response to comment-2:
> > Concerns about deployment (memory and computation costs).
>
> Thank you for your constructive comments. Based on our testing, the model operates smoothly on a standard workstation equipped with 16 GB RAM and a single NVIDIA RTX 3090 GPU. These hardware requirements are highly accessible for clinical deployment.
> ***
> ### Response to comment-3:
> > Suggestions about baselines (prototype-based methods and recent memory-free AD models).
>
> Thank you for your valuable suggestions.
> - ***Prototype-based method***: We expanded our comparisons to include CIPL [1], a recent SOTA ***prototype-based*** interpretability method (TMI 2025). The results (%) validate that our method consistently outperforms CIPL, often by substantial margins (e.g., achieving 91.27% Spe@100%R on JSIEC vs. CIPL's 78.60%, an improvement of over 12%).
>
>     |Dataset|Method|Spe@100%R|Spe@99%R|Spe@95%R|CSR|AUROC|Average Precision|
>     |:-:|:-:|:-:|:-:|:-:|:-:|:-:|:-:|
>     |JSIEC|**Ours**|**91.27**|**91.62**|**94.71**|**88.95**|**98.06**|**96.10**|
>     ||CIPL|78.60|79.57|87.33|73.41|94.83|91.36|
>     |RIADD|**Ours**|**55.35**|**59.39**|**72.92**|**54.38**|**91.32**|**60.35**|
>     ||CIPL|43.99|51.40|64.92|44.85|87.09|59.08|
>     |CheXpert|**Ours**|**68.72**|**74.26**|**84.04**|**70.59**|**96.72**|**94.71**|
>     ||CIPL|43.83|48.72|68.09|47.29|92.57|84.60|
>     |Derm12345|**Ours**|**78.49**|**85.78**|**90.29**|**77.92**|**97.43**|**34.23**|
>     ||CIPL|54.95|68.08|82.04|54.74|94.00|13.37|
>
> - ***Recent memory-free AD model***: Our original manuscript ***already includes SCRD4AD (ICLR 2025)***, which utilizes reverse distillation without memory. To prevent any reader confusion, we will highlight these specific baseline categories more explicitly in the revised manuscript.
>
> [1] Cross-and intra-image prototypical learning for multi-label disease diagnosis and interpretation. TMI'25.
> ***
> ### Response to comment-4:
> > Questions about the scale of dual knowledge banks (number of vectors, memory footprint).
>
> Thank you for your valuable question. The number of vectors and memory footprints for each modality are listed below:
> ||Number of vectors|Memory|
> |:-:|:-:|:-:|
> |Ophthalmology|256k|1,002MB|
> |Radiology|1,024k|4,004MB|
> |Dermatology|196k|766MB|
>
> The hardware requirements are highly accessible, as discussed in *Response to comment-2*.
> ***
> ### Response to comment-5:
> > Questions about the robustness of the pathological bank and denoising strategies.
>
> Thank you for your insightful question. In *Response to comment-1*, we discussed that our method is inherently robust to the normal tissue contamination in the pathological bank, supported by two primary mechanisms. To empirically validate this, we present two additional experiments below.
>
> - ***Stress test***: To validate the robustness of the pathological bank, we designed a ***stress test*** in which we manually injected 20% normal samples into the pathological bank during construction.
> - ***Denoising strategy***: Following you suggestions, we attempted to incorporate a SoftPatch-like ***denoising strategy*** during knowledge bank construction.
>
> As shown in the results below, our model maintains stable performance under stress testing on the JSIEC dataset, and explicit denoising yields comparable scores.
> This confirms that our design is robust to patch-level noise.
>
> ||Original|Stress test|Denoising strategy|
> |:-:|:-:|:-:|:-:|
> |AUROC (%)|98.06|97.71|96.90|
> |Spe@99%R (%)|91.62|90.10|92.11|
> ***
> Again, we sincerely appreciate your time and effort in reviewing. Revisions following your comments have greatly strengthened our manuscript. If you have further questions, please do not hesitate to let us know.
> ***

---

> > ### Author Rebuttal · Reviewer_Zae3 · 2026-04-03
> >
> > I am satisfied with the rebuttal, and I'm gonna keep my score as weak accept.

---

> > > ### Author Response · Authors · 2026-04-03
> > >
> > > Dear Reviewer Zae3,
> > >
> > > Thank you sincerely for taking the time to carefully read our rebuttal and for acknowledging that your concerns have been addressed. We are ***truly encouraged*** by your positive assessment of our work!
> > >
> > > We are glad that our responses were satisfactory. Your insightful comments have ***genuinely helped us improve the quality*** of the paper, and we are grateful for the constructive feedback throughout this process.
> > >
> > > If, upon reflection, you feel that the rebuttal has resolved your concerns, we would be grateful if you ***might consider*** whether a score adjustment could be warranted. Of course, we respect and appreciate your judgment either way.
> > >
> > > Thank you again for your constructive and thoughtful review throughout this process.
> > >
> > > Best regards,
> > >
> > > The Authors

---

### Official Review · Reviewer_4hQu · 2026-03-16

**Soundness:** 2
**Presentation:** 3
**Significance:** 2
**Originality:** 2
**Overall Recommendation:** 3
**Confidence:** 3

**Summary:**

In this paper, the authors proposed EviScreen, a framework for disease screening based on comparisons with historical cases. Specifically, EviScreen constructs and utilizes dual knowledge banks, one storing features of normal cases and the other storing features of pathological cases, where the features are computed using a vision foundation model. Given an input image, EviScreen uses the same vision foundation model to extract input features, and retrieves similar features from both knowledge banks to construct the evidence for the prediction. During prediction, EviScreen performs cross-attention between the input features and evidence features to obtain evidence-aware input features for predicting whether the input image is a normal case or a pathological case. Alternatively, EviScreen also supports a training-free variant called contrastive retrieval, where two distance maps from the input features to those in the normal and pathological knowledge banks are constructed and subtracted to yield an abnormality map, indicating where the pathology is likely located on the input image (if present). In experiments, the authors evaluated the AUROC, specificity at 95%, 99%, and 100% recall, and clear separation rate of EviScreen against competing methods across three medical domains, and show that their EviScreen achieved superior performance to competing methods.

**Compliance With Llm Reviewing Policy:**

Affirmed.

**Ethical Review Concerns:**

N/A.

**Final Justification:**

I am maintaining my original rating. While the authors did address some of my concerns, there are still major setbacks of the proposed method such as the lack of support for multiclass classification.

**Key Questions For Authors:**

From my understanding of reading the paper, the authors trained and evaluated their models on different datasets across the three medical domains (ophthalmology, radiology, and dermatology) to simulate real-life scenarios. What criteria did you use when deciding which datasets should be used for training and validation, and which for testing?

**Limitations:**

Yes, the authors have adequately discussed the limitations and potential negative societal impact of their work.

**Strengths And Weaknesses:**

Strengths:
- The paper presents an interesting case study of using knowledge banks to perform disease screening based on medical images.

Weaknesses:
- The proposed method resembles prototype-based methods (e.g., ProtoPNet (Chen et al., 2019)), which also compares an input image with historical (prototypical) cases in the feature space to make a final prediction. Specifically, prototype-based methods also provide "retrospection interpretability" by comparing similar image patches to prototypical patches (which are analogous to small "knowledge banks") and "localization interpretability" through prototype activation maps. They have been used in medical applications (e.g., Barnett et al., 2021). However, the authors did not discuss these methods, and claimed that "no existing medical image-based disease screening approach provides visual evidence from historical cases to explain its predictions" which is not true.
- While the focus of the paper is on interpretability, the proposed transformer-based evidence-aware reasoning model is hardly interpretable in terms of its model design (cross attention + self-attention).
- Table 1 (the main results table) does not include the AUROC, specificity at 95%, 99%, and 100% recall, and clear separation rate based solely on the abnormality map from contrastive retrieval.
- Average precision should also be reported. It measures precision for various recall levels using different thresholds. This is a harder metric than AUROC, especially in the case of imbalanced datasets.
- The proposed EviScreen does not support multiclass classification (e.g., predicting a particular abnormality category instead of abnormality versus normal).

---

> ### Author Rebuttal · Authors · 2026-03-31
>
> We sincerely appreciate your constructive feedback and the positive recognition of our work as "an interesting case study". We have carefully incorporated your insightful feedback into our revision. Below, we address your concerns point by point.
> ***
> ### Response to comment-1:
> > Concerns about discussing prototype-based methods.
>
> Thank you for highlighting the important perspective of prototype-based methods. We have refined our claims to accurately reflect their contributions and clarified the ***distinct advantages*** of our dual knowledge banks over prototypes:
>
> - ***Computational scalability.*** Prototype-based models require ***computationally intensive training*** to learn prototypes. In contrast, our dual knowledge banks are constructed ***training-free*** using features from a frozen foundation model.
> - ***Capacity scalability.*** Prototype models suffer from ***capacity bottlenecks***. For instance, Figure 6 of CIPL (a SOTA prototype-based model published in TMI 2025 [1]) shows a performance ***decline*** when the number of prototypes per class exceeds 50. Conversely, Figure 5(b) of our manuscript shows our method can scale to ***over 200k vectors*** (10,000 samples).
> - ***Notable performance improvements.*** Benefiting from this scalability, our method preserves a broader range of visual patterns, resulting in significant performance gains. The table below validates that our method consistently outperforms CIPL [1], often by substantial margins (e.g., achieving 91.27% Spe@100%R on JSIEC vs. CIPL's 78.60%, an improvement of over 12%).
>
> |Dataset|Method|Spe@100%R|Spe@99%R|Spe@95%R|CSR|AUROC|Average Precision|
> |:-:|:-:|:-:|:-:|:-:|:-:|:-:|:-:|
> |JSIEC|**Ours**|**91.27**|**91.62**|**94.71**|**88.95**|**98.06**|**96.10**|
> ||CIPL|78.60|79.57|87.33|73.41|94.83|91.36|
> |RIADD|**Ours**|**55.35**|**59.39**|**72.92**|**54.38**|**91.32**|**60.35**|
> ||CIPL|43.99|51.40|64.92|44.85|87.09|59.08|
> |CheXpert|**Ours**|**68.72**|**74.26**|**84.04**|**70.59**|**96.72**|**94.71**|
> ||CIPL|43.83|48.72|68.09|47.29|92.57|84.60|
> |Derm12345|**Ours**|**78.49**|**85.78**|**90.29**|**77.92**|**97.43**|**34.23**|
> ||CIPL|54.95|68.08|82.04|54.74|94.00|13.37|
>
> [1] Cross-and intra-image prototypical learning for multi-label disease diagnosis and interpretation. TMI'25.
> ***
> ### Response to comment-2:
> > Concerns about interpretability of the transformer-based architecture.
>
> Thank you for this helpful comment. The interpretability of our design does not derive from the attention mechanism or model activations. Instead, it stems from the visual evidence from historical cases and the abnormality map.
> ***
> ### Response to comment-3:
> > Suggestions to include results of the training-free variant in Table 1.
>
> Thank you for your helpful advice. While we presented the training-free variant (based on contrastive retrieval) in Figure 5(a) to compare with training-free methods, we agree that including these results directly in Table 1 would improve experimental completeness. We will ***update Table 1 accordingly*** in the revised version.
> ***
> ### Response to comment-4:
> > Suggestions to include average precision.
>
> Thank you for this constructive advice. As shown below, our model consistently outperforms all comparative methods regarding ***average precision (%)***:
>
> |Dataset|Ours|FM|PatchCore*|PatchCore|NFM-DRA|DRA|SCRD4AD|EDC|SimpleNet|CIPL|
> |:-:|:-:|:-:|:-:|:-:|:-:|:-:|:-:|:-:|:-:|:-:|
> |JSIEC|**96.10**|94.24|89.61|86.62|93.23|89.53|89.85|71.44|57.66|91.36|
> |RIADD|**60.35**|38.16|59.73|32.46|40.97|50.45|39.94|11.67|13.44|59.08|
> |CheXpert|**94.71**|91.81|38.15|30.61|34.23|70.64|27.74|33.88|29.90|84.60|
> |Derm12345|**34.23**|28.41|3.24|4.49|5.69|19.70|3.80|2.79|4.22|13.37|
>
> We will ***update Table 1*** in the revised version.
> ***
> ### Response to comment-5:
> > Concerns about multi-class classification.
>
> Thank you for this observation. We would like to clarify that the ***primary scope*** of this clinical-driven research is ***disease screening***, which serves as a clinically vital and distinct step prior to ***disease diagnosis***. Screening aims to identify individuals with abnormalities in an apparently healthy population [2] and is usually formulated as a binary classification task (normal vs. abnormal). Nevertheless, our core mechanisms, such as dual knowledge banks and contrastive retrieval, are highly adaptable and can generalize to multi-class classification.
>
> [2] Principles and practice of screening for disease. World Health Organization. 1968.
> ***
> ### Response to comment-6:
> > Questions about the criteria for deciding training/validation and testing datasets.
>
> We carefully selected datasets with ***reliable annotations and diverse abnormalities*** for testing, and other high-quality datasets for training/validation.
>
> ***
> Thank you again for reviewing our manuscript. The resulting revisions have greatly strengthened our manuscript. If you have further questions, please do not hesitate to let us know.
> ***

---

> > ### Author Rebuttal · Reviewer_4hQu · 2026-04-03
> >
> > The proposed framework is designed for disease screening (binary classification), and cannot easily extend to multiclass classification. The authors did state that the method could be generalized to multi-class classification, but did not offer details regarding how this could be done.
> >
> > I believe that this is a major setback of the method that cannot be easily addressed in a short rebuttal.

---

> > > ### Author Response · Authors · 2026-04-03
> > >
> > > Dear Reviewer 4hQu,
> > >
> > > Thank you sincerely for your thoughtful engagement with our rebuttal and for the time you have dedicated to reviewing our manuscript. We understand that your remaining concerns center on the ***extension of our method to multi-class classification***, rather than the validity of our ***current contribution to disease screening***. We would like to respectfully provide further clarification on this point.
> > >
> > > We fully understand and acknowledge your perspective. However, we would like to emphasize that disease screening and disease diagnosis are ***distinct clinical tasks*** with different objectives, workflows, and real-world requirements. Disease screening, as defined by the World Health Organization (1968), aims to identify individuals with potential abnormalities from an apparently healthy population.
> > >
> > > This binary formulation is ***not a limitation*** of our method, but rather a ***deliberate and clinically oriented design choice*** that directly addresses an important unmet need in medical AI. Accordingly, the evaluation metrics, experimental benchmarks, and the overall framework of this paper are ***purposefully designed around the screening paradigm***. Our results demonstrate strong and consistent performance across multiple datasets and clinically meaningful metrics (e.g., Spe@100%R, AUROC, Average Precision), validating the effectiveness of our approach within its intended scope.
> > >
> > > While our core mechanisms are highly adaptable and hold strong potential to generalize to multi-class classification, this extension is ***not encompassed within the primary contribution of the current work***. Nevertheless, we are genuinely enthusiastic about this direction and view it as a valuable direction for future research. We sincerely hope that the reviewer can consider the contributions of this work within its ***intended clinical scope***.
> > >
> > > We are grateful for this constructive discussion and truly hope our clarifications address your remaining concerns. We warmly welcome any further comments.
> > >
> > > Sincerely,
> > >
> > > Paper 20396 authors

---

### Decision · Program_Chairs · 2026-04-30

**Decision:**

Accept (regular)

**Comment:**

This paper proposes a retrieval  framework for interpretable disease screening from medical images. The method builds dual knowledge banks of regional features from normal and pathological historical cases. It generates evidence maps by contrasting distances to the two knowledge banks, providing interpretability. The framework is evaluated across 10 datasets spanning ophthalmology, radiology, and dermatology and shows good performance, without training on a task. It provides inferential evidence by citing historical cases, which not only improves accuracy but also enhances clinical trust. Still, reviewers were not enthusiastic about the paper with two weak accepts and one weak reject. Criticism was focussed on lack of comparison to prototype methods and speed, memory consumption and scaling of maintaining the knowledge banks. Therefore, I suggest **reject**.